# Can we have it all? On the Trade-off between Spatial and Adversarial Robustness of Neural Networks

**Sandesh Kamath**
Indian Institute of Technology, Hyderabad

**Amit Deshpande**
Microsoft Research India

**K V Subrahmanyam**
Chennai Mathematical Institue, Chennai

**Vineeth N. Balasubramanian**
Indian Institute of Technology, Hyderabad

## Abstract

(Non-)robustness of neural networks to small, adversarial pixel-wise perturbations, and as more recently shown, to even random spatial transformations (e.g., translations, rotations) entreats both theoretical and empirical understanding. Spatial robustness to random translations and rotations is commonly attained via equivariant models (e.g., StdCNNs, GCNNs) and training augmentation, whereas adversarial robustness is typically achieved by adversarial training. In this paper, we prove a quantitative trade-off between spatial and adversarial robustness in a simple statistical setting. We complement this empirically by showing that: (a) as the spatial robustness of equivariant models improves by training augmentation with progressively larger transformations, their adversarial robustness worsens progressively, and (b) as the state-of-the-art robust models are adversarially trained with progressively larger pixel-wise perturbations, their spatial robustness drops progressively. Towards achieving Pareto-optimality in this trade-off, we propose a method based on curriculum learning that trains gradually on more difficult perturbations (both spatial and adversarial) to improve spatial and adversarial robustness simultaneously.

## 1 Introduction

Neural network (NN) models have achieved state-of-the-art performance on several image tasks over the last few years. However, they have been known to be vulnerable to adversarial attacks that make small, imperceptible pixel-wise perturbations to images [39, 5]. Most pixel-wise adversarial perturbations studied in previous work are carefully constructed, model-dependent and, in most cases also input-dependent, perturbations of small $\ell_\infty$ norm (and sometimes $\ell_1$ or $\ell_2$ norm). To achieve adversarial robustness, a popular, principled, and effective defense against $\ell_\infty$ adversarial attacks known at present is adversarial training [17, 29, 37, 49].

From another perspective, recent work has also shown natural, physical-world attacks on neural network models using 2D and 3D spatial transformations such as translations, rotations and pose [47, 14, 42, 1]. Even random spatial transformations such as random translations and random rotations degrade the accuracy of neural network models considerably [12]. Recent benchmarks also measure the robustness of NN models to the average-case, model-agnostic perturbations (e.g., noise, blur, brightness, contrast) or natural distribution shifts, and not only to the worst-case adversarial perturbations [21, 40]. Spatial robustness or robustness to spatial transformations (e.g., translations, rotations) has been long studied as a problem of model invariance, and is generally addressed using equivariant models or training augmentation. While standard Convolutional Neural Networks (StdCNNs) are translation-equivariant, recent efforts have resulted in equivariant NN models for other transformations such as rotation, flip [16, 8, 10, 45, 31, 26, 27, 13], and scaling [30, 46, 38].

35th Conference on Neural Information Processing Systems (NeurIPS 2021).

Such group-equivariant NN models (GCNNs) are designed to be invariant to a specific group of transformations, so they need training augmentation to attain spatial robustness to random or average-case transformations. However, aided by clever weight-sharing, these recent works have achieved good spatial robustness performance with lesser training augmentation than their non-equivariant counterparts.

In this paper, we address an inevitable trade-off between the average-case spatial robustness and the worst-case $\ell_\infty$ adversarial robustness in neural network models, and show results theoretically as well as empirically. On the one hand, as we attain better spatial robustness via equivariant models and larger training augmentation, the adversarial robustness worsens. On the other hand, as we attain better adversarial robustness via adversarial training against larger pixel-wise perturbations, the spatial robustness worsens. We then provide a curriculum learning-based approach to mitigate this issue and achieve a certain degree of Pareto-optimality in this trade-off.

Our work extends recent progress on two important questions, namely, the robustness to multiple attacks and the robustness vs accuracy trade-off. The robustness to translations (up to $\pm3$ pixels) and rotations (up to $\pm30°$) has been shown to have a fundamentally different loss landscape than the robustness to adversarial $\ell_p$ perturbations [12]. A trade-off between robustness to adversarial $\ell_\infty$ perturbations and adversarial translations and rotations up to $\pm3$ pixels and $\pm40°$, respectively, was also recently noted [42]. Adversarial attacks on $\ell_p$ adversarially robust models by exploiting invariance at intermediate layers have also been of interest in recent work [23, 41]. A key difference of our work with these efforts is that: (i) we study spatial robustness to *random* translations and rotations (which is more closely related to practice when training NN models and has not been studied in these earlier efforts), instead of adversarial translations or rotations; (ii) we study spatial robustness w.r.t transformations over their entire range, and do not restrict ourselves to a $\pm30 - 40°$ range, for instance; and (iii) we consider the use of group-equivariant models with training augmentation (which are designed for spatial robustness) in our studies, beyond just StdCNN models. Our results show a *progressive decline* in spatial robustness as adversarial robustness improves, and vice versa – a trend that was not explicitly noted in previous results. Our result can also be interpreted as a progressive trade-off between adversarial robustness and generalization accuracy itself [43, 49, 48], where generalization performance measures accuracy on a test distribution that includes natural data augmentation such as random translations, rotations, etc.

Given the nature of this trade-off, we then ask the question of how one can obtain a model close to the Pareto-optimal frontier for adversarial and random spatial robustness. Are there training strategies that can help achieve a balance between both? We observe that the obvious solutions of training in two stages – first, adversarial training and then training augmentation by spatial transformations, or vice versa – does not help address this issue. This results in a version of catastrophic forgetting, where as the model learns robustness on one of these fronts, it loses robustness on the other. We instead show that a curriculum learning-based strategy, where the model is trained by applying an adversarial perturbation to a randomly transformed input with gradually increased difficulty levels, gives solutions consistently closer to the Pareto-optimal frontier across well-known datasets and models. This strategy reflects the progressive nature noted of the trade-off itself, and highlights the importance of the observation of this trend.

Our key contributions can be summarized as follows:

- We study a trade-off between adversarial robustness and spatial robustness (with a focus on *random* spatial transformations) both theoretically and empirically, on a setting closer to natural neural network training, which to the best of our knowledge, has not been considered before. We explain the trade-off between spatial and adversarial robustness with a well-motivated theoretical construction.

- We conduct a comprehensive suite of experiments across popularly used models and datasets, and find that models trained for spatial robustness (StdCNNs and GCNNs) progressively improve spatial robustness when their training is augmented with larger random spatial transformations but while doing so, their $\ell_\infty$ adversarial robustness drops progressively. Similarly, models adversarially trained using progressively larger $\ell_\infty$-norm attacks improve their adversarial robustness but while doing so, their spatial robustness drops progressively.

- We provide a simple yet effective approach based on curriculum learning that gives better-balanced, Pareto-optimal spatial and adversarial robustness compared to several natural baselines derived from combining adversarial training and training augmentation by random spatial transformations.

## 2 Problem Setting and Formulation

We begin by describing our problem formally. Let $(X, Y)$ denote a random input-label pair from a given data distribution over the input-label pairs $(x, y) \in \mathcal{X} \times \mathcal{Y}$. Let $f$ be a neural network classifier. The accuracy of $f$ is given by $\Pr(f(X) = Y)$, the fraction of inputs where the prediction matches the true label. For $\epsilon > 0$, an $\ell_\infty$ adversarial attack $\mathcal{A}$ maps each input $x$ to $x + \mathcal{A}(x)$ such that $\|\mathcal{A}(x)\|_\infty \leq \epsilon$, for all $x \in \mathcal{X}$. The *fooling rate* of $\mathcal{A}$ is given by $\Pr(f(X + \mathcal{A}(X)) \neq f(X))$, the fraction of inputs where the prediction changes after adversarial perturbation. For a given spatial transformation $T$, the classifier $f$ is said to be $T$-invariant if the predicted label remains unchanged after the transformation $T$ for all inputs; in other words, $f(Tx) = f(x)$, for all $x \in \mathcal{X}$. The spatial transformations such as translation, rotation and flip preserve the $\ell_\infty$ norm, so $T\mathcal{A}(x)$ is an adversarial perturbation of small $\ell_\infty$ norm for the input $Tx$ for a $T$-invariant classifier. Therefore, for a given perturbation bound $\epsilon > 0$, the maximum fooling rate for any $T$-invariant classifier $f$ on the transformed data $\{Tx : x \in \mathcal{X}\}$ must be equal to the maximum fooling rate for $f$ on the original data $\mathcal{X}$. It is a bit more subtle when $f$ is not truly invariant, that is, $f(Tx) = f(x)$ for most inputs $x$ but not all $x$. We define the rate of invariance of a classifier $f$ to a transformation $T$ as $\Pr(f(TX) = f(X))$ or the fraction of test images whose predicted labels remain unchanged under the transformation $T$. For a class of transformations, e.g., random rotation $r$ from the range $[-\theta^\circ, +\theta^\circ]$, we define the *rate of invariance* as the average rate of invariance over transformations $T$ in this class, i.e., $\Pr(f(r(X)) = f(X))$, where the probability is over the random input $X$ as well as the random transformation $r$. The rate of invariance is $100\%$ if the model $f$ is truly invariant. When $f$ is not truly invariant, the interplay between the invariance under transformations and robustness under adversarial perturbations of small $\ell_\infty$-norm is subtle. *This interplay is exactly what we investigate.*

In this paper, we study neural network models and the simultaneous interplay between their spatial robustness, viz, rate of invariance for spatial transformations (such as random rotations between $[-\theta^\circ, +\theta^\circ]$ for $\theta$ varying in the range $[0, 180]$), and their adversarial robustness to pixel-wise perturbations of $\ell_\infty$ norm at most $\epsilon$. Measuring the robustness of a model to adversarial perturbations of $\ell_p$ norm at most $\epsilon$ is NP-hard [24, 37]. Previous work on the comparison of different adversarial attacks has shown the Projected Gradient Descent (PGD) attack to be among the strongest [2]. Hence, we use PGD-based adversarial training as the methodology for adversarial robustness in a model, as commonly done. We show that our conclusions also hold for other forms of adversarial training such as TRADES [49], which gives a trade-off between adversarial robustness and natural accuracy (on the original unperturbed data).

Theoretically speaking, we prove a quantitative trade-off between spatial and adversarial robustness in a simple, natural statistical setting proposed in previous work [43]. Our theoretical results (described in Sec 3) show that there is a trade-off between spatial and adversarial robustness, and that it is not always possible to have both in a given model. Empirically, we study the following: (a) change in $\ell_\infty$ adversarial robustness as we improve only the rate of spatial robustness using training augmentation with progressively larger transformations; (b) change in spatial invariance as we improve only adversarial robustness using PGD adversarial training with progressively larger $\ell_\infty$-norm of pixel-wise perturbations. We study StdCNNs, group-equivariant GCNNs, as well as popular architectures used to assess adversarial robustness [29, 49]. Our empirical studies are conducted on MNIST, CIFAR10, CIFAR100 as well as Tiny ImageNet datasets, thus showing the general nature of these results (described in Sec 4). Importantly, as stated earlier, we consider random spatial transformations that are more commonplace than adversarial rotations in previous work [42]. We look at the entire possible range for spatial transformations. Similarly, we normalize the underlying dataset, and compute the accuracy of a given model on test inputs adversarially perturbed using PGD attack of $\ell_\infty$ norm at most $\epsilon$, for $\epsilon$ varying over $[0, 1]$. In contrast with the certifiable lower bounds on robustness [3, 15, 35, 28, 32], we study upper bounds on spatial and adversarial robustness, respectively.

## 3 Spatial-Adversarial Robustness Trade-off

In this section, we prove the trade-off between spatial and adversarial robustness theoretically, and support this result with experiments in Sec 4. We use $\mathcal{A}(x)$ to denote an adversarial $\ell_\infty$ perturbation and $r(x)$ to denote a random spatial transformation. Equivariant model constructions often consider a group of transformations and construct a neural network model invariant to this group. For simplicity, we consider a cyclic group that can model a rotation group (e.g., integer multiples of $30°$), or horizontal/vertical translations (e.g., horizontal translations by multiples of, say, $\pm 4$ pixels).

We take a construction proposed in previous work for a robustness vs accuracy trade-off [43], and infuse it with a simple idea from the theory of error correcting codes [34]. Consider a binary cyclic code of length $d$, where each codeword is a binary string in $\{-1, 1\}^d$ and all the codewords can be obtained by applying successive cyclic shift of coordinates to a generator codeword; see Chapter 8 in [34]. The cyclic code is said to have relative distance $\delta$, if any two codewords differ in at least $\delta d$ coordinates. Let $c = (c_1, c_2, \ldots, c_d)$ be the generator codeword. Consider a random input-label pair $(X, Y)$, with $X = (X_0, X_1, \ldots, X_d)$ taking values in $\mathbb{R}^{d+1}$ and $Y$ taking values in $\{-1, 1\}$, generated as follows. The class label $Y$ takes value $\pm 1$ with probability $1/2$ each. $X_0 \mid Y = y$ takes value $y$ with probability $p$ and $-y$ with probability $1 - p$, for some $p \geq 1/2$. The remaining coordinates are independent and normally distributed with $X_t \mid Y = y$ as $N(2c_t y/\sqrt{d}, 1)$, for $1 \leq t \leq d$. Let $m$ be an integer divisor of $d$, and let $r_j(x)$ denote the cyclic shift of $mj$ coordinates applied to $(x_1, x_2, \ldots, x_d)$ while keeping $x_0$ unchanged, i.e., $r_j(x) = (x_0, x_{mj+1}, x_{mj+2}, \ldots, x_d, x_1, \ldots, x_{mj})$. For example, a 90° rotation can be considered as a cyclic permutation of pixels with $m = d/4$ and the center pixel $x_0$ being fixed. Now let $r(x)$ denote a random permutation that takes value $r_j(x)$, uniformly at random over $j \in \{1, 2, \ldots, d/m\}$. For example, using $m = d/4$ as above, $r(x)$ can model a random multiple of 90° rotation.

First, we show that achieving a high degree of spatial robustness on the above distribution is non-trivial even if we have high accuracy on the original data without any transformations. So our subsequent trade-off between spatial and adversarial robustness cannot be derived from previously known trade-offs between accuracy and adversarial robustness [43, 49].

**Proposition 1.** *There exists $p \geq 1/2$ and a cyclic code with relative distance $\delta \geq 3/8$ such that given the input distribution defined as above, the classifier of maximum accuracy on input $(X, Y)$ has accuracy at least $97\%$. Similarly, the classifier of maximum accuracy on the transformed input $(r_j(X), Y)$ also has accuracy at least $97\%$. However, when the classifier of maximum accuracy on $(X, Y)$ is applied to $(r_j(X), Y)$, for any $j$, it has accuracy at most $85\%$.*

Next we show that if we have a classifier of high adversarial robustness on the above data distribution, then it must have low spatial robustness and vice versa. The proof of this follows from a previously known accuracy vs robustness trade-off by Tsipras et al.[43]. However, their distribution is invariant to any permutation of the coordinates $x_1, x_2, \ldots, x_d$, so the accuracy and the spatial robustness are equal for their distribution.

**Theorem 2.** *Given the input distribution defined as above, any $\eta > 0$ and any classifier $f : \mathbb{R}^{d+1} \to \{-1, 1\}$, if the adversarial accuracy of $f$ is at least $1 - \eta$, then the spatial accuracy of $f$ is at most $\dfrac{\eta \, p}{(1 - p)}$. Similarly, if the spatial accuracy of $f$ is at least $1 - \eta$ then the adversarial accuracy $f$ is at most $1 - \dfrac{(1 - p)(1 - \eta)}{p}$.*

The above results show that adversarial and spatial robustness can not just be high simultaneously. We do not explicitly model the effect of adversarial training with larger $\ell_\infty$ perturbations but, as a reasonable proxy, consider the increase in the adversarial accuracy for a fixed perturbation bound. A tighter analysis of Theorem 2 towards a gradual trade-off between spatial and adversarial robustness is presented in the supplementary.

## 4 Empirical Analysis

**Experimental Setup.** In order to study the effect of spatial transformations (e.g. rotations, translations) and adversarial perturbations (e.g. Projected Gradient Descent or PGD attacks), we use well-known popularly used NN architectures as well as state-of-the-art networks known to have strong invariance [8], as well as networks which are known to have strong robustness [29, 49]. We study the spatial vs adversarial robustness trade-off using well-known architectures (described below) on MNIST, CIFAR10, CIFAR100 and Tiny ImageNet datasets. Our code is made available for reproducibility.

*Spatially Robust Model Architectures:* StdCNNs are known to be translation-equivariant by design, and GCNNs [8] are rotation-equivariant by design through clever weight sharing [25]. Equivariant models, especially GCNNs, when trained with random rotation augmentations have been observed to come very close to being truly rotation-invariant [8, 9, 7] (or spatially robust in our context). We

hence use both StdCNNs and equivalent GCNNs trained with suitable data augmentations for our studies with spatially robust architectures. In particular, for each StdCNN we use, the corresponding GCNN architecture is obtained by replacing the layer operations with equivalent GCNN operations as in [8][1]. For the StdCNN, we use a Conv-Conv-Pool-Conv-Conv-Pool-FC-FC architecture for MNIST (more details in the supplementary); VGG16 [36] and ResNet18 [19] for CIFAR10 and CIFAR100; and ResNet18 for the Tiny ImageNet dataset.

*Adversarially Robust Model Architectures:* For adversarial training, we use a LeNet-based architecture for MNIST[2] and a ResNet-based architecture for CIFAR10[3]. Both these models are exactly as given in [29]. For CIFAR100, we use the popularly used WideResNet-34[4] architecture also used in [49]. We use ResNet18 [19] for the Tiny ImageNet dataset.

*Training Data Augmentation: Spatial*: (a) **Aug - R** : Data is augmented with random rotations in the range $\pm\theta°$ given $\theta$, along with random crops and random horizontal flips (for MNIST alone, we do not apply crop and horizontal flips); (b) **Aug - T**: Data is augmented with random translations within $[-i, +i]$ range of pixels in the image, given $i$ (eg. for CIFAR10 with $i = 0.1$ is $32 * 0.1 \approx \pm 3px$) in both horizontal and vertical directions; (c) **Aug - RT**: Data is augmented with random rotations in $\pm i * 180°$ and random translations within $[-i, +i]$ range of pixels in the image (eg. for CIFAR10 with $i = 0.1$ is $0.1 * 180° = \pm 18°$ rotation and $32 * 0.1 \approx \pm 3px$ translation), here no cropping and no horizontal flip is used. We use nearest neighbour interpolation and black filling to obtain the transformed image. *Adversarial* : **Adv - PGD**: Adversarial training using PGD-perturbed adversarial samples using an $\epsilon$-budget of given $\epsilon$. Our experiments with PGD use a random start, 40 iterations, and step size 0.01 on MNIST, and a random start, 10 iterations, and step size $2/255$ on CIFAR10, CIFAR100 and Tiny ImageNet. While we study the trade-off under a fixed PGD setting our trend holds even with different PGD hyperparameter settings. Refer supplementary J for details. Our results, presented in the next section, are best understood by noting the augmentation method mentioned in the figure caption. For example, in Fig 1(a), the augmentation scheme used is **Aug-R**. The red line (annotated as 60 in the legend) corresponds to the model trained with random rotation augmentations in the range $\pm 60°$.

*Hardware Configuration:* We used a server with 4 Nvidia GeForce GTX 1080i GPU to run all the experiments in the paper.

*Evaluation Metrics:* We quantify performance using a spatial invariance profile and an adversarial robustness profile.

*Spatial invariance:* We quantify rotation invariance by measuring the rate of invariance or the fraction of test images whose predicted label remains the same after rotation by a random angle between $[-\theta°, \theta°]$. As $\theta$ varies from 0 to 180, we plot the rate of invariance. We call this the *rotation invariance profile* of a given model. Similarly, by training a model with other augmentation schemes given above, we obtain a *translation invariance profile* and a *rotation-translation invariance profile* for a given model.

*Adversarial robustness:* Similarly, we quantify the $\ell_\infty$ adversarial robustness of a given model to a fixed adversarial attack (e.g., PGD) and a fixed $\ell_\infty$ norm $\epsilon \in [0, 1]$ by (1 - fooling rate), i.e., the fraction of test inputs for which their predicted label does not change after adversarial perturbation. We plot this for $\epsilon$ varying from 0 to 1. We call this the *robustness profile* of a given model. (One can also choose accuracy instead of 1 - fooling rate and we observe similar trends, as shown in the supplementary. We use 1 - fooling rate for clearer visual presentation of our main results.)

**Results.** We now present our results on spatially robust model architectures followed by adversarially robust ones.

*Results on Spatially Robust Model Architectures:* Fig 1 present our results on the trade-off on adversarial robustness for VGG16, trained with spatial augmentations, on CIFAR10. (More results including ResNet18 on CIFAR10, VGG16/ResNet18 on CIFAR100 and the StdCNN/GCNN-based model on MNIST are deferred to the supplementary owing to space constraints.)

---

[1]https://github.com/adambielski/pytorch-gconv-experiments

[2]https://github.com/MadryLab/mnist_challenge/

[3]https://github.com/MadryLab/cifar10_challenge

[4]https://github.com/yaodongyu/TRADES/models/

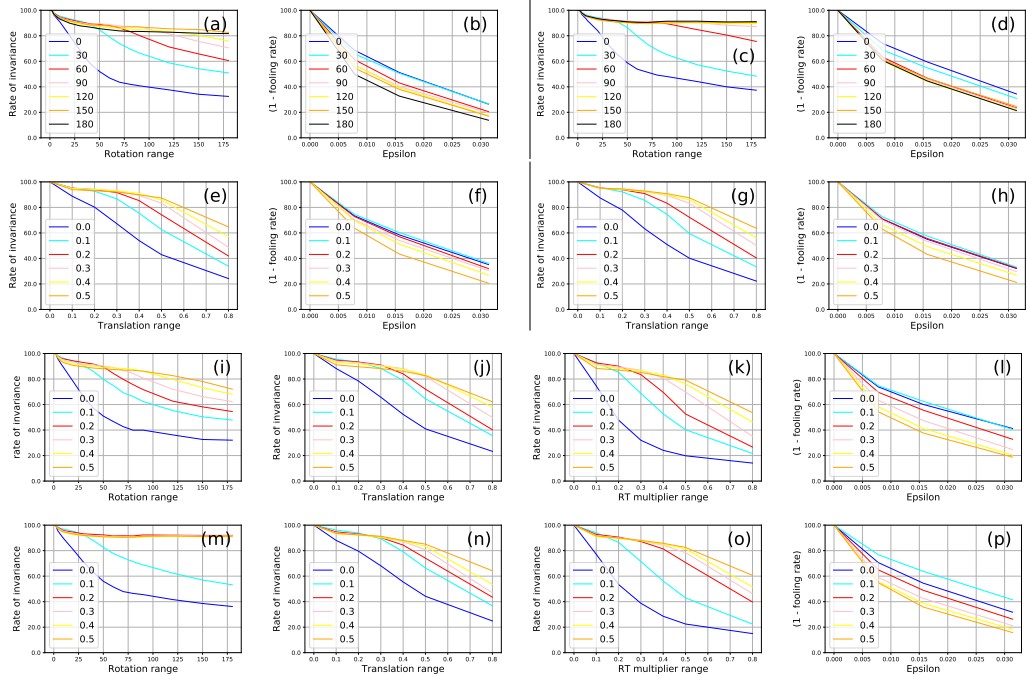

Figure 1: On CIFAR10, For VGG16 model (a-b) **Aug - R** StdCNN, (c-d) **Aug - R** GCNN, (e-f) **Aug - T** StdCNN, (g-h) **Aug - T** GCNN, (i-l) **Aug - RT** StdCNN, (m-p) **Aug - RT**, invariance profiles of StdCNN/GCNN models and corresponding robustness profiles.

For any fixed $\theta \in [0, 180]$, we take an equivariant model, namely, StdCNN or GCNN, and augment its training data by random rotations from $[-\theta°, +\theta°]$. For example, Fig 1(b) show how the robustness profile of StdCNN changes on CIFAR10, as we increase the degree $\theta$ used in training augmentation of the model. We use PGD attack provided in versions of AdverTorch [11] to obtain the robustness profiles. Similarly, Figs 1(a) show the rotation invariance profile of the same models on CIFAR10. The black line in Fig 1(b) shows that the adversarial robustness of a StdCNN which is trained to handle rotations up to $\pm 180$ degrees on CIFAR10, drops by more than 50%, even when the $\epsilon$ budget for PGD attack is only $2/255$. Similarly, the black line in Fig 1(a) shows this model's rotation invariance profile - this model is invariant to larger rotations on test data. This can be contrasted with the model depicted by the red line - this StdCNN is trained to handle rotations up to $60$ degrees. The rotation invariance profile of this model is below that of the model depicted by the black line and hence is lesser invariant to large rotations. However, this model can handle adversarial $\ell_\infty$-perturbations up to $2/255$ on unrotated data, with an accuracy more than $10\%$, as seen from the red line in Fig 1(b). The remaining plots in Figs 1 show similar trends for translation, as well as a combination of rotation and translation (e.g. plots (e-p) in Figs 1 for CIFAR10) for both StdCNN and GCNN models (e.g. Figs 1c,d).

The above results show that *the spatial robustness of these models improves by training augmentation but at the cost of their adversarial robustness, indicating a trade-off between spatial and adversarial robustness.* This trade-off exists in both StdCNN and GCNN models.

*Results on Adversarially Robust Model Architectures:* Fig 2 presents our results on the trade-off on spatial robustness for models trained adversarially using PGD [29] on MNIST, CIFAR10 and CIFAR100 datasets. Each row corresponds to experiments for a single dataset with a plot of robustness profile (for which they are trained) followed by the spatial invariance profiles. Each colored line in the plots corresponds to a model adversarially trained with a different value of an $\epsilon$-budget.

On MNIST, adversarial training with PGD with larger $\epsilon$ results in a drop in the invariance profile of the LeNet-based model; in Fig 2(b-d), the red line (PGD with $\epsilon = 0.3$) is below the light blue line (PGD with $\epsilon = 0.1$). Similar observations hold for the ResNet model on CIFAR10 (see Fig 2 (f-h)), as well as for WideResNet-34 on CIFAR100 (see Fig 2(j-l)). To complete this picture, the robustness profile curves confirm that as these models are trained with PGD using larger $\epsilon$ budget,

their adversarial robustness increases. The robustness profile curves of the LeNet model trained with a larger PGD budget dominates the robustness profile curve of the same model trained with a smaller PGD budget; the red line in Fig 2(a) dominates the light blue line. This is true of the ResNet model too, as in Figs 2(e) and 2(i).

In other words, *adversarial training with progressively larger $\epsilon$ leads to a drop in the rate of spatial invariance on test data.* Fig 3 shows an additional result with StdCNN (VGG-16) with PGD-based adversarial training on CIFAR10. These results too support the same findings.

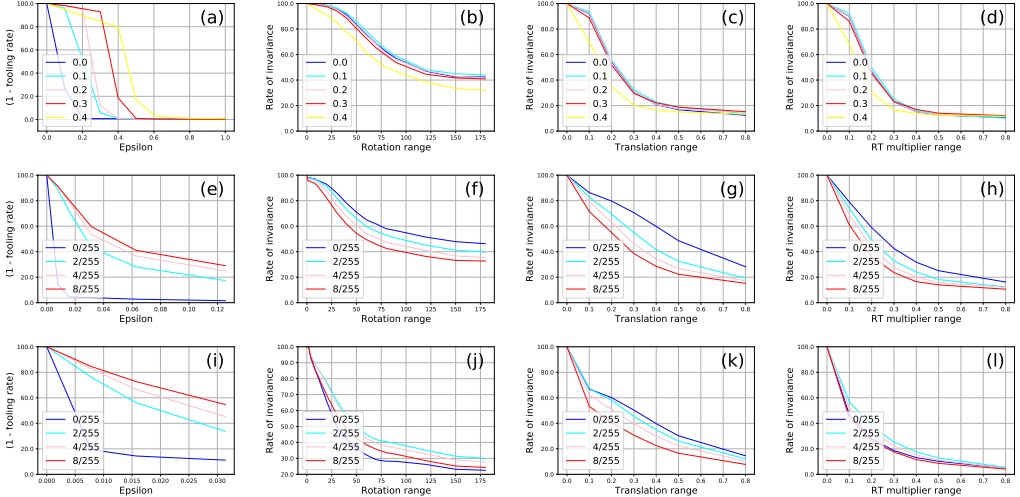

Figure 2: **Adv - PGD**, (1) Robustness, (2) Rotation, (3) Translation, (4) Rotation-Translation invariance profiles for PGD adversarially trained models. (a-d) LeNet based model from [29] on MNIST (e-h) ResNet based model from [29] on CIFAR10, (i-l) WideResNet34 model on CIFAR100. Different colored lines represent models adversarially trained with different $\ell_\infty$ budgets $\epsilon \in [0, 1]$.

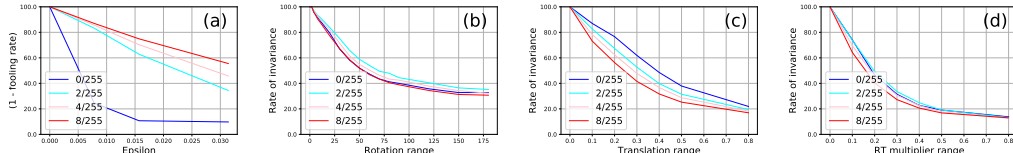

Figure 3: On CIFAR10, **Adv - PGD**, For StdCNN/VGG16 model (a) Robustness profile, (b) Rotation invariance profile, (c) Translation invariance profile (d) Rotation-Translation invariance profile. Different colored lines represent models adversarially trained with different $\ell_\infty$ budgets $\epsilon \in [0, 1]$.

*Results on Tiny ImageNet:* Appendix I studies both adversarial robustness of spatially robust models, as well as spatial invariance of adversarially robust models on the Tiny ImageNet dataset. Our observations mentioned so far holds here too, corroborating our claim that this trade-off is not merely theoretical but noticed in real-world datasets too.

*Going beyond PGD-based Adversarial Training:* While we have used the popular PGD-based adversarial training for the study of robust models in all the results so far, we conduct experimental studies with another recent popular adversarial method, TRADES [49]. Fig 4 shows the results of these studies on CIFAR10, where we observe the same qualitative behavior of the models.

*Additional Empirical Evidence (Average Perturbation Distance to Boundary) :* While more results and implementation details are provided in the supplementary section, we briefly provide an interesting evidence to our claims based on the average distance to the decision boundary (details in Appendix L). We notice that there is a distinct reduction in the distance to decision boundary as model achieves better spatial invariance (Figure 5). This reduction in distance can be linked to the drop in adversarial robustness. Such a study also has the potential to be used to study trade-offs in the presence of other common corruptions [21, 22, 20].

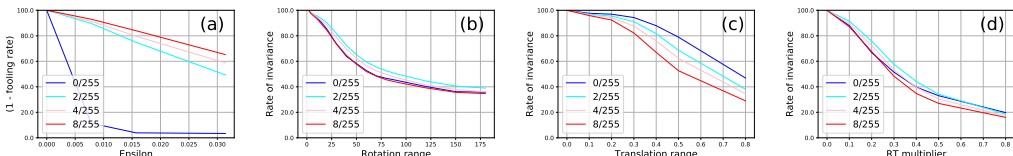

Figure 4: On CIFAR10, **Adv - TRADES**, For StdCNN/WideResNet34 model (a) Robustness , (b) Rotation invariance , (c) Translation invariance (d) Rotation-Translation invariance profiles. Different colored lines represent models adversarially trained with different $\ell_\infty$ budgets $\epsilon \in [0, 1]$.

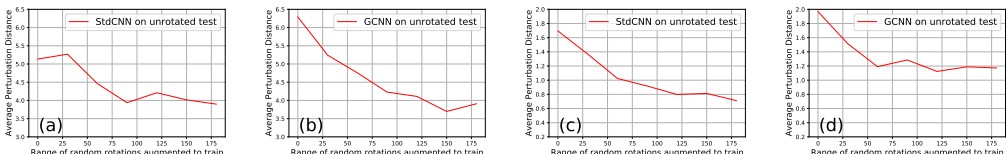

Figure 5: Average perturbation distance of StdCNN/GCNN on (a-b) MNIST(based on appendix Table 2), (c-d) CIFAR10(VGG16) wrt unrotated test data.

## 5    Curriculum Learning-based Approach for Pareto-Optimality

The progressive nature of the trade-off between spatial invariance and adversarial robustness elicits the need to study training methods that can simultaneously achieve robustness on both of these fronts. In this section, we propose a curriculum learning-based training strategy that consistently provides models close to the Pareto-optimal frontier of the trade-off. An obvious strategy to obtain both kinds of robustness would be to sequentially combine adversarial training and augmentation by spatial transformations, i.e., take an adversarially trained network and re-train it with spatial (e.g., rotation) augmentation, or vice versa (Appendix K). We however observe that such a strategy presents a version of catastrophic forgetting where the robustness learned earlier is forgotten when the other robustness is enforced. A model thus trained has either good spatial robustness or good adversarial robustness but not both. We hence explore a curriculum learning strategy to this end.

Curriculum learning is inherently inspired by how humans learn; first learning simple tasks and then followed by gradually difficult tasks [4]. Previous work has demonstrated its effectiveness in training DNN models [44, 18]. A curriculum-based adversarial training has also been proposed [6], which however focuses only on adversarial robustness. For simultaneous spatial and adversarial robustness, we propose CuSP (**Cu**rriculum based **S**patial-Adversarial Robustness training for **P**areto-Optimality), where the difficulty of spatial augmentation and adversarial strength is gradually increased each time the learning rate is updated as part of a schedule. In particular, each time the learning rate is updated, we increase the spatial augmentation range $[-\theta°, \theta°]$ as well as the adversarial perturbation bound $\epsilon$ and adversarial train on a randomly perturbed input within that range. This strategy reflects an optimization procedure where the difficulty of robustness handled by the model is increased when the model treads more slowly (lower learning rate) along areas of the loss surface. The outline of the proposed CuSP approach (**Cu**rriculum based **S**patial-Adversarial Robustness training for **P**areto-Optimality) is given in Algorithm 1.

To study the goodness of the proposed strategy, we look at three metrics: (a) spatial accuracy on large random transformations $\Pr\left(f(T_{180}(X)) = Y\right)$, where $T_{180}$ denotes a random rotation in the range $[-180°, 180°]$, (b) adversarial accuracy at large perturbations $\Pr\left(f(X + A_{8/255}(X)) = Y\right)$, where $A_{8/255}$ denotes an adversarial attack (PGD or TRADES) of $\ell_\infty$-norm bounded by $\epsilon = 8/255$, and finally, (c) natural accuracy on the unperturbed data $\Pr\left(f(X) = Y\right)$. The last one is included for completeness because we want better spatial and adversarial robustness simultaneously but not lose on natural accuracy in the bargain [43, 49].

*Results:* We studied CuSP across models and datasets in our experiments, and report the results of VGG16 on CIFAR10 in Table 1. The remainder of these results are presented in Appendix M. We follow the learning rate schedule 75-90-100 as used in TRADES [49] where the learning rate starting

**Algorithm 1:** CuSP: **Cu**rriculum-based **S**patial-Adversarial Robustness Training for **P**areto-Optimality

**Input** : Training data $(X, Y)$

**Parameters** : No. of epochs: K, Learning rate: $\eta$, Adversarial training (AT) method: PGD or
TRADES, Range of spatial transformations: $\theta_1 \leq \theta_2 \leq \ldots \leq \theta_l$,
Adversarial perturbation bounds: $\epsilon_1 \leq \epsilon_2 \leq \ldots \leq \epsilon_s$,
Learning rate schedule: $\eta_1 > \eta_2 > \ldots > \eta_t$

**Output** : Model parameters $\phi$

1 Initialize model parameters $\phi$; $i, j, k \leftarrow 1$
2 $\theta \leftarrow \theta_i, \epsilon \leftarrow \epsilon_j, \eta \leftarrow \eta_k$
3 **for** *epochs* $t = 1, \ldots, E$ **do**
4   **if** $k \leftarrow k+1$; $\eta \leftarrow \eta_k$ /* Change in learning rate schedule       */
5   **then** $i \leftarrow i + 1$; $j \leftarrow j + 1$; Update $\theta \leftarrow \theta_i$ and $\epsilon \leftarrow \epsilon_j$;
6   $\phi \leftarrow AdversarialTraining_\phi(T_\theta(X) + A_\epsilon(T_\theta(X), \eta))$
   /* Minimize loss on input randomly transformed within $[-\theta, \theta]$ and an
    adversarial attack of norm at most $\epsilon$            */
7 **end**

| | Training Method | Adv (PGD) Accuracy(%) | Std Accuracy(%) | Spatial Accuracy(%) |
|---|---|---|---|---|
| 1 | Natural (Aug 0) | 00.05±0.06 | **92.89±0.34** | 33.99±0.54 |
| 2 | Natural (Aug 180) | 00.00±0.00 | **85.00±1.36** | **83.78±0.95** |
| 3 | PGD (Aug 0) | **44.88±0.22** | **80.08±0.58** | 33.07±0.81 |
| 4 | PGD (Aug 180) | 32.79±0.73 | 54.60±0.51 | 53.69±0.48 |
| 5 | PGD (Aug 0) → Aug 180 | 00.00±0.00 | **85.46±0.25** | **83.99±0.34** |
| 6 | Aug 180 → PGD (Aug 0) | **45.80±1.05** | **81.03±0.14** | 33.19±0.33 |
| 7 | PGD (Aug 0) → PGD (Aug 180) | 35.19±0.31 | 58.96±0.21 | 57.84±0.15 |
| 8 | Aug 180 → PGD (Aug 180) | 35.94±0.35 | 60.27±0.39 | 59.02±0.21 |
| 9 | CuSP (PGD, 30-60-180, $\{\frac{2}{255}, \frac{4}{255}, \frac{8}{255}\}$) | **38.63±0.11** | 65.92±0.35 | 53.31±0.13 |
| 10 | CuSP (PGD, 60-120-180, $\{\frac{2}{255}, \frac{4}{255}, \frac{8}{255}\}$) | **37.18±0.14** | 65.88±0.11 | **59.09±0.18** |
| 11 | CuSP (PGD, 120-150-180, $\{\frac{2}{255}, \frac{4}{255}, \frac{8}{255}\}$) | 36.01±0.36 | 64.96±0.10 | **61.41±0.61** |
| 12 | CuSP (PGD, 180, $\{\frac{2}{255}, \frac{4}{255}, \frac{8}{255}\}$) | 33.63±0.38 | 62.17±0.50 | **61.07±0.43** |
| 13 | CuSP (PGD, 120-150-180,$\{\frac{8}{255}\}$) | 35.57±0.16 | 58.08±0.26 | 55.54±0.19 |
| 14 | PGD (Aug 0) → CuSP (PGD, 120-150-180, $\{\frac{2}{255}, \frac{4}{255}, \frac{8}{255}\}$) | **36.92±0.28** | **66.25±0.68** | **62.51±0.12** |
| 15 | Aug 180 → CuSP (PGD, 120-150-180, $\{\frac{2}{255}, \frac{4}{255}, \frac{8}{255}\}$) | **37.16±0.12** | **66.58±0.12** | **63.04±0.11** |

Table 1: Comparison of performance of proposed CuSP for CIFAR10 on StdCNN/VGG16 with other baseline strategies. (Aug $\theta$): denotes training data augmented with random rotations in the range $[-\theta, +\theta]$; $a \to b$: denotes $a$ sequentially followed by $b$ during training. The top 7 values in each column are in boldface.

with 0.1 decays by a factor of 0.1 successively at the 75-th, 90-th and 100-th epochs. Our algorithm works with given sets of values for $\theta$ and $\epsilon$ and does not optimize for the choice of these values. We empirically studied various combinations of $(\theta, \epsilon)$ values as $\theta$ varies over $\{0, 30, 60°, \ldots, 180\}$ and $\epsilon$ varies over $\{0, 2/255, 4/255, 8/255\}$. While each of these options improves performance over the baselines, we found the list of $\theta = \{120°, 150°, 180°\}$ to give the best results on this particular setting. We also include results for CuSP training after a warm start, viz., PGD with no augmentation and natural training with $[-180°, +180°]$ augmentation. Figure 6 compares the performance of CuSP against other baseline training strategies, visualized in the context of the Pareto frontier for PGD accuracy and spatial accuracy.

## 6 Conclusions and Future Work

In this work, we present a comprehensive discussion on the trade-off between spatial and adversarial robustness in neural network models. While a few recent efforts have studied similar trade-offs, we study random spatial transformations in this work, which are more popularly used in practice than any adversarial spatial transformations. We study the trade-off both theoretically and empirically, and

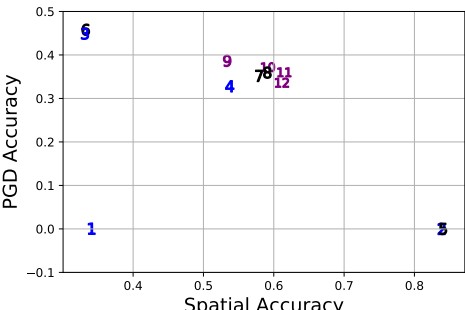
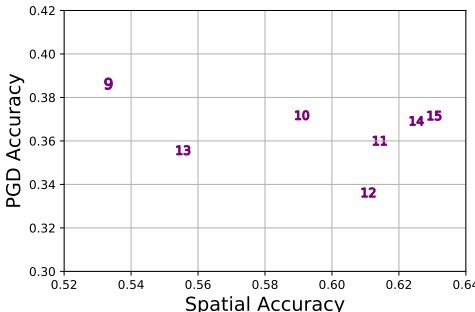

Figure 6: Visualization of performance of CuSP based on PGD against other baseline strategies on CIFAR10 for StdCNN/VGG16 model (each index corresponds to a row in Table 1). On the left, we compare CuSP against various baselines. On the right, we zoom in to compare different variants of CuSP.

prove a quantitative trade-off between spatial and adversarial robustness in a simple statistical setting used in earlier work. Our experiments show that as equivariant models (StdCNNs and GCNNs) are trained with progressively larger spatial transformations, their spatial invariance improves but at the cost of their adversarial robustness. Similarly, adversarial training with perturbations of progressively increasing norms improves the robustness of well-known neural network models, but with a resulting drop in their spatial invariance. We also provide a curriculum learning-based training method to obtain a solution that is close to Pareto-optimal in comparison to other baseline adversarial training methods. While CuSP gradually increases both the range of spatial transformation $\theta$ in the augmentation and the adversarial perturbation bound $\epsilon$, we conjecture that tuning these further with the learning rate schedule along with a warm start for model parameters could give further improvements. Recently, [21, 22, 20] have shown that study of other common corruptions is important in the study of state-of-the-art NN models. Similar studies between spatial invariance and such common corruptions is an interesting direction of future work.

**Broader Impact.** Adversarial robustness is one of the most important problems in better understanding the functioning of neural networks. Another important problem is how networks can efficiently handle various spatial transformations like rotation, translation, etc. The trade-off studied and the algorithm proposed to find a better Pareto-optimal in this paper tries to understand the interaction between the two which is a natural question. Hence, we believe this study is towards better understanding of the networks learning capability. There is no known detrimental social impact of our work.

**Acknowledgements and Funding Transparency Statement.** Sandesh Kamath would like to thank Microsoft Research India for funding a part of this work through his postdoctoral research fellowship at IIT Hyderabad.

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
