## Supplementary Material : Can we have it all? On the Trade-off between Spatial and Adversarial Robustness of Neural Networks

The supplementary material contains proofs, additional experiments to show that our results continue to hold across different datasets and different models, and additional empirical evidence for the spatial vs adversarial robustness trade-off that we could not fully include in the main paper owing to space constraints.

## A   Proof of Proposition 1

This section provides the proof for Proposition 1. The parameter $\delta$ below is mostly for illustration and not optimized. We consider $d$ to be large, as the small adversarial perturbation we consider has $\ell_\infty$ norm $\leq 4/\sqrt{d}$.

**Proposition 1.** *There exists $p \geq 1/2$ and a cyclic code with relative distance $\delta \geq 3/8$ such that given the input distribution defined as above, the classifier of maximum accuracy on input $(X, Y)$ has accuracy at least 97%. Similarly, the classifier of maximum accuracy on the transformed input $(r_j(X), Y)$ also has accuracy at least 97%. However, when the classifier of maximum accuracy on $(X, Y)$ is applied to $(r_j(X), Y)$, for any $j$, it has accuracy at most 85%.*

*Proof.* Let $\rho(x_0)$ be equal to $p/(1-p)$, if $x_0 = 1$, and $(1-p)/p$, if $x_0 = -1$. The Bayes classifier $f^*$ that maximizes the accuracy is:

$$
\begin{aligned}
f^*(x) &= \text{sign}\left(\frac{\Pr(Y=1, X=x)}{\Pr(Y=-1, X=x)} - 1\right) \\
&= \text{sign}\left(\rho(x_0) \frac{\prod_{t=1}^d \exp\left(\frac{-(x_t - 2c_t/\sqrt{d})^2}{2}\right)}{\prod_{t=1}^d \exp\left(\frac{-(x_t + 2c_t/\sqrt{d})^2}{2}\right)} - 1\right) \\
&= \text{sign}\left(\frac{\prod_{t=1}^d \exp\left(\frac{-(x_t - 2c_t/\sqrt{d})^2}{2}\right)}{\prod_{t=1}^d \exp\left(\frac{-(x_t + 2c_t/\sqrt{d})^2}{2}\right)} - \frac{1}{\rho(x_0)}\right) \\
&= \text{sign}\left(\sum_{t=1}^d c_t x_t + \frac{\sqrt{d}}{4} \log \rho(x_0)\right),
\end{aligned}
$$

because the quadratic terms and the constants cancel out. Note that the above classifier of maximum accuracy can be represented as a single decision tree node at $x_0$ followed by a linear classifier in the remaining coordinates $x_1, x_2, \ldots, x_d$. Moreover, $c_t = \pm 1$ and $X_t \mid Y = y$ is normally distributed as $N(2c_t y/\sqrt{d}, 1)$, so $\sum_{t=1}^d c_t X_t \mid Y = y$ is normally distributed as $Z_y = N(2y\sqrt{d}, d)$. As $p \to 1/2$, we have $\log \rho(x_0) \to 0$. It is known that $\Pr(Z \geq \mu - 2\sigma) = \Pr(Z \leq \mu + 2\sigma) \approx 0.977$, for any normally distributed $Z$ with mean $\mu$ and standard deviation $\sigma$. So the $\text{sign}(Z_y)$ matches $y$ with probability at least 97%, and the accuracy of $f^*$ is at least 97%.

Similarly, the classifier $f_j^*$ that maximizes the accuracy on the transformed data $(\tilde{X}, Y) = (r_j(X), Y)$ is given by

$$
f_j^*(\tilde{x}) = \text{sign}\left(\sum_{t=1}^d (r_j(c))_t \tilde{x}_t + \frac{\sqrt{d}}{4} \log \rho(\tilde{x}_0)\right),
$$

and the same proof mutatis mutandis works to show that its accuracy on $(\tilde{X}, Y)$ is at least 97%.

Let $S_j$ be the subset of coordinates $t$ with $(r_j(c))_t = c_t$. Since our code is binary, we get $(r_j(c))_t = -c_t$, for $t \notin S_j$. Therefore,

$$
f_j^*(x) = \text{sign}\left(\sum_{t \in S_j} c_t x_t - \sum_{t \notin S_j} c_t x_t + \frac{\sqrt{d}}{4} \log \rho(x_0)\right).
$$

Since $c_t = \pm 1$ and $X_t \mid Y = y$ is normally distributed as $N(2c_t y/\sqrt{d}, 1)$, we have $c_t X_t \mid Y = y$ normally distributed as $N(2y/\sqrt{d}, 1)$. Moreover, we get that $\sum_{t \in S_j} c_t X_t - \sum_{t \notin S_j} c_t X_t \mid Y = y$ is normally distributed as $N(2y(2\,|S_j| - d)/\sqrt{d}, 2\,|S_j| - d)$. Note that $|S_j| \leq (1 - \delta)d$. Thus, for $\delta \geq 3/8$ we get

$$\frac{|2y(2\,|S_j| - d)|}{\sqrt{2\,|S_j| - d}} \leq 2\sqrt{1 - 2\delta} = 1.$$

For the existence of a cyclic code with relative distance $\delta \geq 3/8$, see Chapter 8 of [34]. It is known that $\Pr\left(Z \geq \mu - \sigma\right) = \Pr\left(Z \leq \mu + \sigma\right) \approx 0.841$, for any normally distributed $Z$ with mean $\mu$ and standard deviation $\sigma$. Using $p$ sufficiently close to $1/2$, we get that $\Pr\left(f_j^*(X) = Y\right)$ is at most $85\%$. Since $r(x)$ is a random transformation $r_j(x)$ where $j$ is picked uniformly at random from $\{1, 2, \ldots, d/m\}$, the same bound holds for the data transformed using $r(x)$. $\qquad\square$

## B   Proof of Theorem 2

We now prove Theorem 2 on the trade-off between spatial and adversarial robustness.

**Theorem 2.** *Given the input distribution defined as above, any $\eta > 0$ and a classifier $f : \mathbb{R}^{d+1} \to \{-1, 1\}$, if the adversarial accuracy is at least $1 - \eta$, then the spatial accuracy is at most $\dfrac{\eta\,p}{(1 - p)}$. Similarly, if the spatial accuracy is at least $1 - \eta$ then the adversarial accuracy is at most $1 - \dfrac{(1 - p)(1 - \eta)}{p}$.*

*Proof.* Observe that $r_j(X)$ only permutes the coordinates of $X$. Hence,

$$\min_{\mathcal{A}} \Pr(f(X + \mathcal{A}(X)) = Y) \leq \Pr(f(r_j(X) + \mathcal{A}(r_j(X))) = Y) \qquad (1)$$

The known adversarial robustness vs. standard accuracy trade-off proof [43] implies, for all $j$,

$$\Pr(f(r_j(X) + \mathcal{A}(r_j(X))) = Y) \leq 1 - \frac{1 - p}{p}\,\Pr(f(r_j(X)) = Y).$$

By averaging the above over all $j$, we get

$$\min_{\mathcal{A}} \Pr(f(X + \mathcal{A}(X)) = Y) \leq 1 - \frac{m}{d}\,\frac{1 - p}{p}\sum_{j=1}^{d/m} \Pr(f(r_j(X)) = Y)$$

$$= 1 - \frac{1 - p}{p}\,\Pr(f(r(X) = Y).$$

Now if the adversarial robustness (i.e., the LHS above) is at least $1 - \eta$, then the spatial robustness can be upper bounded as

$$\Pr(f(r(X)) = Y) \leq \frac{\eta\,p}{1 - p}. \qquad (2)$$

The other side of the trade-off can be proved similarly. If the spatial robustness $\Pr(f(r(X) = Y)$ is at least $1 - \eta$ then there exists at least one index $j$ such that $\Pr(f(r_j(X)) = Y) \geq 1 - \eta$. So again using the above adversarial robustness vs. standard accuracy trade-off for that particular index $j$ we get

$$\Pr(f(r_j(X) + \mathcal{A}(r_j(X))) = Y) \leq 1 - \frac{1 - p}{p}\,\Pr(f(r_j(X)) = Y)$$

$$\leq 1 - \frac{(1 - p)(1 - \eta)}{p}.$$

Hence, the adversarial robustness can be upper bounded as follows, using the observation that $r_j(X)$ only permutes the coordinates of $X$.

$$\min_{\mathcal{A}} \Pr(f(X + \mathcal{A}(X)) = Y) \leq \Pr(f(r_j(X) + \mathcal{A}(r_j(X))) = Y)$$

$$\leq 1 - \frac{(1 - p)(1 - \eta)}{p},$$

$\qquad\square$

# C   Tighter Analysis and a Modified Version of Theorem 2

We now give a tighter analysis and prove a modified version of Theorem 2 towards exhibiting a trade-off between spatial and adversarial robustness. It would be interesting to close the gap between our theoretical trade-off and the progressive or gradual trade-off observed in our experiments.

**Theorem 3.** *(Theorem 2, modified) Given the input distribution defined as above and a classifier* $f : \mathbb{R}^{d+1} \to \{-1, 1\}$, *suppose* $\Pr\left(f(X + \mathcal{A}(X)) = Y\right) \geq 1 - \eta$, *for all adversarial attacks* $\mathcal{A}$ *of* $\ell_\infty$ *norm bounded by* $4/\sqrt{d}$. *Then*

$$\left(1 - \frac{m}{d}\right)(1 - \eta) \leq \Pr\left(f(r(X)) = Y\right) \leq 1 - \frac{m}{d}\left(1 - \frac{\eta\, p}{1 - p}\right).$$

*Note that as* $p \to 1/2$ *and* $\eta \to 0$, *both the bounds approach* $1 - m/d$, *which decreases as* $m$ *increases. For* $m = d/2$, *this accuracy is as bad as that of a random classifier.*

*Proof.* Let $\mathcal{A}(x)$ denote an adversarial perturbation for $(x, y)$ given by $(\mathcal{A}(x))_0 = 0$, and $(\mathcal{A}(x))_t = -4c_t y/\sqrt{d}$, for $1 \leq t \leq d$. Note that $\|\mathcal{A}(x)\|_\infty = 4/\sqrt{d}$, for all $x \in \mathcal{X}$. Let $G_y$ denotes $(X_1, \ldots, X_d)| Y = y$, the last $d$ coordinates conditioned on $Y = y$. We observe that $X + A(X)| Y = y$ has its last $d$ coordinates distributed as $G_{-y}$, i.e., the adversarial attack that takes $X$ to $X + \mathcal{A}(X)$ turns $G_y$ into $G_{-y}$.

$$\Pr\left(f(X + \mathcal{A}(X)) \neq Y\right)$$

$$= \frac{1}{2} \sum_{y \in \{-1, 1\}} \Pr\left(f((X_0, G_{-y})) = -y\right)$$

$$= \frac{1}{2}\Big\{ \sum_{y \in \{-1, 1\}} p\, \Pr\left(f((y, G_{-y})) = -y\right)$$

$$\qquad\qquad + (1 - p)\, \Pr\left(f((-y, G_{-y})) = -y\right) \Big\}$$

$$\geq \frac{1 - p}{2p}\Big\{ \sum_{y \in \{-1, 1\}} (1 - p)\, \Pr\left(f((y, G_{-y})) = -y\right)$$

$$\qquad\qquad + p\, \Pr\left(f((-y, G_{-y})) = -y\right) \Big\}, \qquad \text{using } p \geq 1/2$$

$$= \frac{1 - p}{2p}\Big\{ \sum_{y \in \{-1, 1\}} (1 - p)\, \Pr\left(f((-y, G_y)) = y\right)$$

$$\qquad\qquad + p\, \Pr\left(f((y, G_y)) = y\right) \Big\}$$

$$= \frac{1 - p}{p}\, \Pr\left(f(X) = Y\right),$$

From the above statement, we get that if the adversarial accuracy $\Pr\left(f(X + \mathcal{A}(X)) = Y\right) \geq 1 - \eta$ then the standard accuracy $\Pr\left(f(X) = Y\right) \leq \eta\, p/(1 - p)$. Using this we get:

$$\Pr\left(f(r(X)) = Y\right)$$

$$= \frac{m}{d} \sum_{j=1}^{d/m} \Pr\left(f(r_j(X) = Y\right)$$

$$= \frac{m}{d}\, \Pr\left(f(X) = Y\right) + \frac{m}{d} \sum_{j=1}^{d/m - 1} \Pr\left(f(r_j(X) = Y\right)$$

$$\leq \frac{m}{d} \cdot \frac{\eta\, p}{1 - p} + \frac{m}{d} \cdot \left(\frac{d}{m} - 1\right)$$

$$= 1 - \frac{m}{d}\left(1 - \frac{\eta\, p}{1 - p}\right).$$

Two important things to note are as follows. First, the above upper bound on the accuracy after random transformation $r$ decreases as $m$ increases. Note that larger rotations mean larger $m$, e.g., a random rotation from $\{0°, 90°, 180°, 270°\}$ can be modeled by $m = d/4$. Second, for $m = d/2$ and $p \to 1/2$, this bound tends to $1/2 + \eta/2$. When $\eta$ is small, this is only slightly better than a random binary classifier.

Observe that the transformation $r_j(x)$ can also be achieved as an adversarial perturbation $x + \mathcal{A}_j(x)$, where $(\mathcal{A}_j(x))_0 = 0$ and $(\mathcal{A}_j(x))_t = (r_j(x))_t - x_t \in \{-4y/\sqrt{d}, 0, 4y/\sqrt{d}\}$, for $1 \leq t \leq d$. Using

this observation, we get

$$\Pr\left(f(r(X)) = Y\right) = \frac{m}{d} \sum_{j=1}^{d/m} \Pr\left(f(r_j(X) = Y\right)$$

$$\geq \frac{m}{d} \sum_{j=1}^{d/m-1} \Pr\left(f(X + \mathcal{A}_j(X) = Y\right)$$

$$\geq \left(1 - \frac{m}{d}\right)(1 - \eta).$$

Note that for $m = d/2$, this bound becomes $(1 - \eta)/2$. When $\eta$ is small, this is only slightly worse than a random binary classifier. In other words, for $m = d/2$, high adversarial robustness implies that spatial robustness must be as bad as that of a random classifier. $\qquad\square$

## D   Choice of Evaluation Metrics: Fooling Rate and Invariance vs Adversarial and Spatial Accuracy

In the main paper, our theoretical results are about the adversarial accuracy $\Pr\left(f(X + \mathcal{A}(X)) = Y\right)$ and the spatial accuracy $\Pr\left(f(r(X)) = Y\right)$. However, the experiments show the fooling rate $\Pr\left(f(X + \mathcal{A}(X)) = f(X)\right)$ and the rate of invariance $\Pr\left(f(r(X)) = f(X)\right)$, respectively. In fact, they are reasonably proxies for each other, respectively, when the standard accuracy $\Pr\left(f(X) = Y\right)$ is high. In Fig 7, we plot the adversarial and the spatial accuracy in (a) and (c), alongside (1 - fooling rate) and rate of invariance in (b) and (d). As evident, our results and claims on the trade-off continue to hold in either case. In the main paper, we present our results using the fooling rate and the rate of invariance because that ensures the same starting point for all the curves, and provides a better visual representation to compare adversarial and spatial robustness of different models.

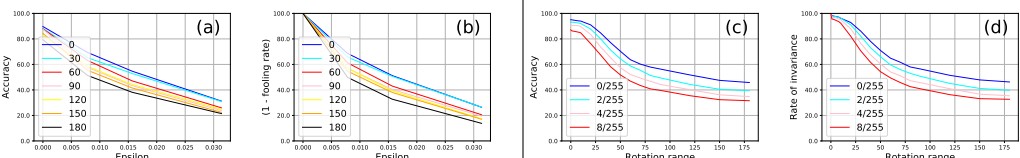

Figure 7: (a-b) Plots of accuracy and (1-fooling rate) for StdCNN/VGG16 trained on CIFAR10; (c-d) Plots of accuracy and rate of invariance for ResNet-based model from [29] adversarially trained on CIFAR10. Note the same starting points on (b) & (d), which allow for better comparison.

## E   Details of Experimental Setup

We describe in detail the settings for all our experiments.

*Datasets:* We use the standard benchmark train-validation-test splits of all the datasets used in this work, that is publicly available. MNIST dataset consists of $70,000$ images of $28 \times 28$ size, divided into 10 classes: $55,000$ used for training, $5,000$ for validation and $10,000$ for testing. CIFAR10 dataset consists of $60,000$ images of $32 \times 32$ size, divided into 10 classes: $40,000$ used for training, $10,000$ for validation and $10,000$ for testing. CIFAR100 dataset consist of $60,000$ images of $32 \times 32$ size, divided into 100 classes: $40,000$ used for training, $10,000$ for validation and $10,000$ for testing. Tiny-ImageNet dataset consists of $100,000$ images of $64 \times 64$ size, divided into 200 classes: $80,000$ used for training, $10,000$ for validation and $10,000$ for testing.

*Spatially Robust Model Architectures:* StdCNNs are known to be translation-equivariant by design, and GCNNs [8] are rotation-equivariant by design through clever weight sharing [25]. Equivariant models, especially GCNNs, when trained with random rotation augmentations have been observed to come very close to being truly rotation-invariant [8, 9, 7] (or spatially robust in our context). We hence use both StdCNNs and equivalent GCNNs trained with suitable data augmentations for our studies with spatially robust architectures. In particular, for each StdCNN we use, the corresponding GCNN architecture is obtained by replacing the layer operations with equivalent GCNN operations

as in [8][5]. For the StdCNN, we use the architecture given in Table 2 for MNIST; VGG16 [36] and ResNet18 [19] for CIFAR10 and CIFAR100; and ResNet18 for the Tiny ImageNet dataset.

*Adversarially Robust Model Architectures:* For adversarial training, we use a LeNet-based architecture for MNIST[6] and a ResNet-based architecture for CIFAR10[7]. Both these models are exactly as given in [29]. For CIFAR100, we use the popularly used WideResNet-34[8] architecture also used in [49]. We use ResNet18 [19] for the Tiny ImageNet dataset.

*Training Data Augmentation: Spatial*: (a) **Aug - R** : Data is augmented with random rotations in the range $\pm\theta°$ given $\theta$, along with random crops and random horizontal flips (for MNIST alone, we do not apply crop and horizontal flips); (b) **Aug - T**: Data is augmented with random translations within $[-i, +i]$ range of pixels in the image, given $i$ (eg. for CIFAR10 with $i = 0.1$ is $32 * 0.1 \approx \pm 3px$) in both horizontal and vertical directions; (c) **Aug - RT**: Data is augmented with random rotations in $\pm i * 180°$ and random translations within $[-i, +i]$ range of pixels in the image (eg. for CIFAR10 with $i = 0.1$ is $0.1 * 180° = \pm 18°$ rotation and $32 * 0.1 \approx \pm 3px$ translation), here no cropping and no horizontal flip is used. Random transformation is picked uniformly at random in the given transformation range, e.g., given $\theta$, rotations are picked uniformly at random from $[-\theta°, +\theta°]$ for augmentation. We use nearest neighbour interpolation and black filling to obtain the transformed image. *Adversarial* : **Adv - PGD**: Adversarial training using PGD-perturbed adversarial samples using an $\epsilon$-budget of given $\epsilon$. Our experiments with PGD use a random start, 40 iterations, and step size 0.01 on MNIST, and a random start, 10 iterations, and step size $2/255$ on CIFAR10, CIFAR100 and Tiny ImageNet. Our results are best understood by noting the augmentation method mentioned in the figure caption. For example, in Fig 1(a), the augmentation scheme used is **Aug-R**. The red line (annotated as 60 in the legend) corresponds to the model trained with random rotation augmentations in the range $\pm 60°$.

*Hyperparameter Settings for CuSP algorithm:* The empirical evaluations of the CuSP algorithm are based on settings used in the TRADES method. For CIFAR10, we train the model for 120 epochs with the learning rate schedule 75-90-100 where the learning rate starting with 0.1 decays by a factor of 0.1 successively at the 75-th, 90-th and 100-th epochs. TRADES is used with $\frac{1}{\lambda} = 5.0$. PGD-based perturbation is used with step size $= \frac{2}{255}$ or 0.007 and number of steps $= 10$ with the appropriate $\epsilon$ budget.

*Hardware Configuration:* We used a computing server with 4 Nvidia GeForce GTX 1080i GPUs to run all experiments in the paper.

*Tool License:* AdverTorch [11] provides the tool under GNU/ LGPL. TRADES and code for [29] are under MIT License.

# F    Experiments on MNIST

Table 2 shows the details of the model architecture used for our experiments on the MNIST dataset. Fig 8 contains the invariance and robustness profiles of StdCNN and GCNN models on MNIST. We observe the same trends noted in the main paper in these results too.

| Standard CNN | GCNN |
|---|---|
| Conv(10,3,3) + Relu | P4ConvZ2(10,3,3) + Relu |
| Conv(10,3,3) + Relu | P4ConvP4(10,3,3) + Relu |
| Max Pooling(2,2) | Group Spatial Max Pooling(2,2) |
| Conv(20,3,3) + Relu | P4ConvP4(20,3,3) + Relu |
| Conv(20,3,3) + Relu | P4ConvP4(20,3,3) + Relu |
| Max Pooling(2,2) | Group Spatial Max Pooling(2,2) |
| FC(50) + Relu | FC(50) + Relu |
| Dropout(0.5) | Dropout(0.5) |
| FC(10) + Softmax | FC(10) + Softmax |

Table 2: Architectures used for MNIST experiments

---

[5]https://github.com/adambielski/pytorch-gconv-experiments
[6]https://github.com/MadryLab/mnist_challenge/
[7]https://github.com/MadryLab/cifar10_challenge
[8]https://github.com/yaodongyu/TRADES/models/

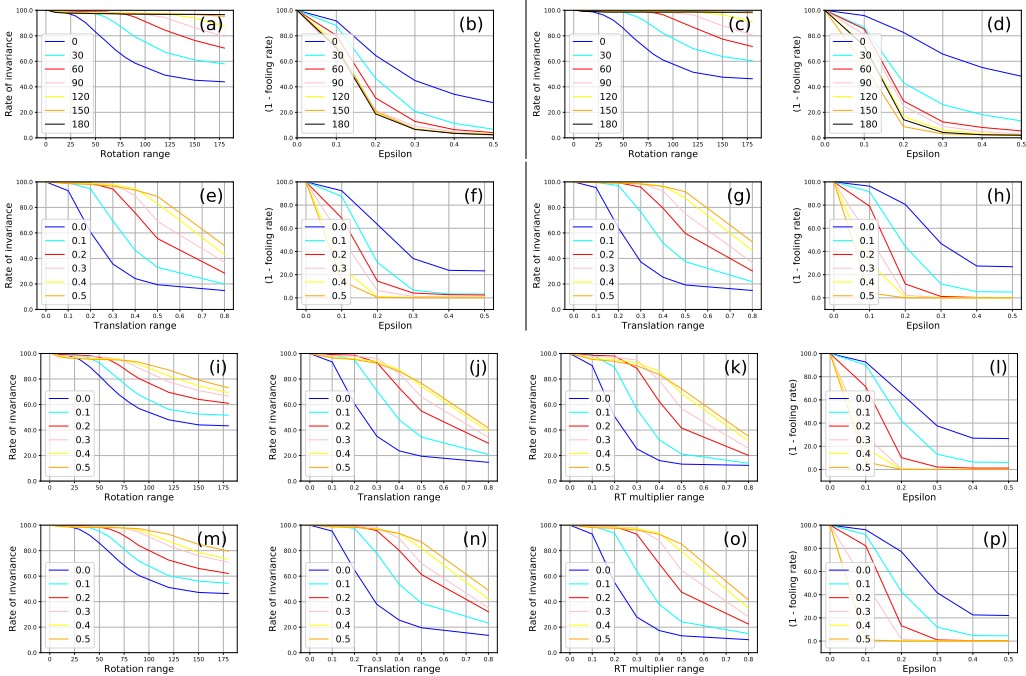

Figure 8: On MNIST, (a-b) **Aug - R** StdCNN, (c-d) **Aug - R** GCNN, (e-f) **Aug - T** StdCNN, (g-h) **Aug - T** GCNN, (i-l) **Aug - RT** StdCNN, (m-p) **Aug - RT** GCNN, invariance profiles of StdCNN/GCNN models and corresponding robustness profiles.

## G   Additional Results on CIFAR10

In the main paper (Fig 1), we presented results for experiments carried out using VGG16 on CIFAR10. Please see Fig 9 for additional experiments on CIFAR10 using the ResNet18 model. These complement the experiments in the main paper and show that our results continue to hold across different models.

## H   Additional Results on CIFAR100

In the main paper, we presented results for experiments carried out using VGG16 on CIFAR10. Please see Figs 10 and 11 for additional experiments on CIFAR100 using the VGG16 and ResNet18 models, respectively. These complement the experiments in the main paper and show that our results continue to hold across different models and datasets.

## I   Additional Results on Tiny-ImageNet

In the main paper, we presented results for experiments carried out using VGG16 on CIFAR10. Please see Figs 12 and 13 for additional experiments on Tiny-ImageNet using the ResNet18 and Densenet121 models, respectively. These complement the experiments in the main paper and show that our results continue to hold across different models and datasets.

## J   Results with different PGD hyperparameter settings

The results in the main paper using PGD attack were with the well known setting of $k = 10$ and step size = $2/255$. We checked the adversarial accuracy for the following 4 variants of PGD attack with $\epsilon = 8/255$ but different number of steps $k$ and step sizes:

  (a)  $k = 10$, step size = $2/255$ or $0.0078$,

  (b)  $k = 20$, step size = $2/255$ or $0.0078$,

  (c)  $k = 100$, step size = $2/255$ or $0.0078$,

  (d)  $k = 100$, step size = $1/255$ or $0.0039$

| Training Method | Adv (PGD) Accuracy(%) | Std Accuracy(%) | Spatial Accuracy(%) (a-b-c-d) |
|---|---|---|---|
| PGD (Aug 0) | 79.60 | 32.51 | 45.05 - 44.75 - 43.26 - 43.27 |
| PGD (Aug 180) | 54.29 | 53.92 | 33.11 - 32.99 - 32.79 - 32.71 |

Table 3: Results similar to Table 1 row indexed 3 (PGD (Aug 0)) and row indexed 4 (PGD (Aug 180)) with different PGD hyperparameter settings on StdCNN/VGG16 with CIFAR10.

In Table 3, we note the results based on the above settings of PGD attack on adversarially trained StdCNN/VGG16 network on CIFAR10 with $0°$ and $\pm180°$ augmented data. We observe that our conclusions about spatial and adversarial robustness trade-off continue to hold for (a-b-c-d) above.

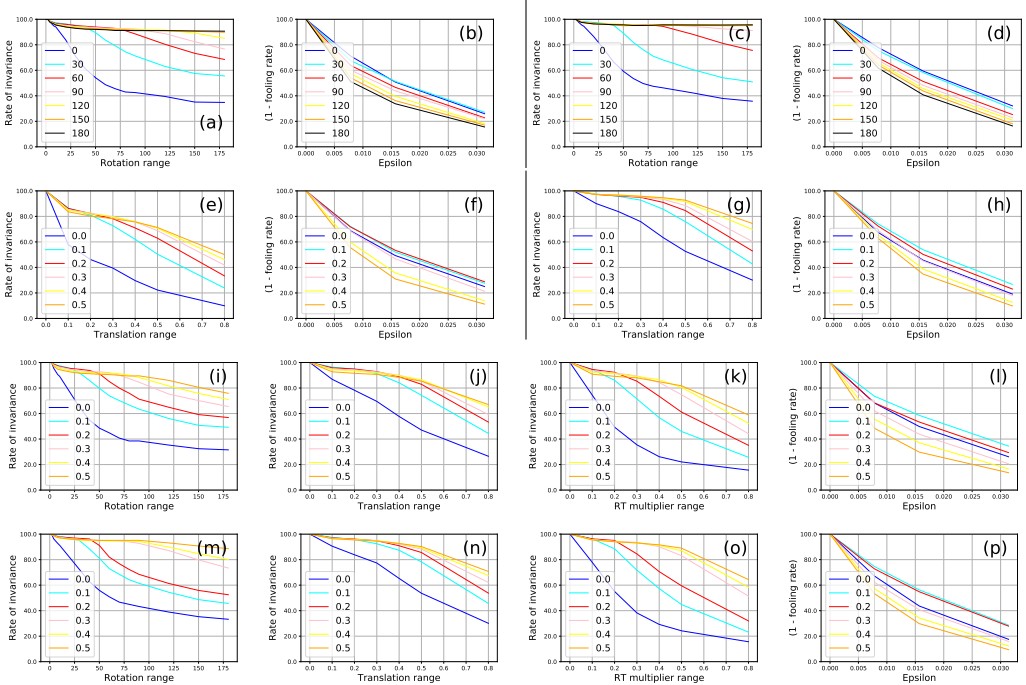

Figure 9: *(Best viewed in color, zoomed in)* On CIFAR10, For ResNet18 model (a-b) **Aug - R** StdCNN, (c-d) **Aug - R** GCNN, (e-f) **Aug - T** StdCNN, (g-h) **Aug - T** GCNN, (i-l) **Aug - RT** StdCNN, (m-p) **Aug - RT**, invariance profiles of StdCNN/GCNN models and corresponding robustness profiles.

# K    Combining Spatial and Adversarial Training

We additionally performed experiments to study training using spatial augmentations as well as adversarial training. The spatial vs adversarial robustness trade-off continues to hold, e.g., the improvement in spatial robustness that we get by spatial transformation augmentation during training is lost when we do adversarial training to improve adversarial robustness. Fig 14 shows translation invariance and $\ell_\infty$ robustness for StdCNN/VGG16 models that are first trained with progressively larger ranges of translation augmentation on CIFAR10, followed by adversarially training them with a fixed $\ell_\infty$ budget of $\epsilon = 8/255$. The results show the same trends as before despite the combined training.

# L    Additional Empirical Evidence: Average Perturbation Distance to the Decision Boundary

For each test image, adversarial attacks find perturbations of the test point with small $\ell_\infty$ norm that would change the prediction of the given model. Most adversarial attacks do so by finding the directions in which the loss function of the model changes the most. In order to explain why these networks become vulnerable to pixel-wise attacks as they learn more rotations, we see how the distance of the test points to the decision boundary changes as the networks learn larger rotations. This is illustrated in Fig 15 where we show the distance of a test point $x_0$ to the boundary $D_0$ (resp. $D_{180}$) when the model is trained with zero (resp. $\pm180°$) rotations. We use the $L_2$ attack

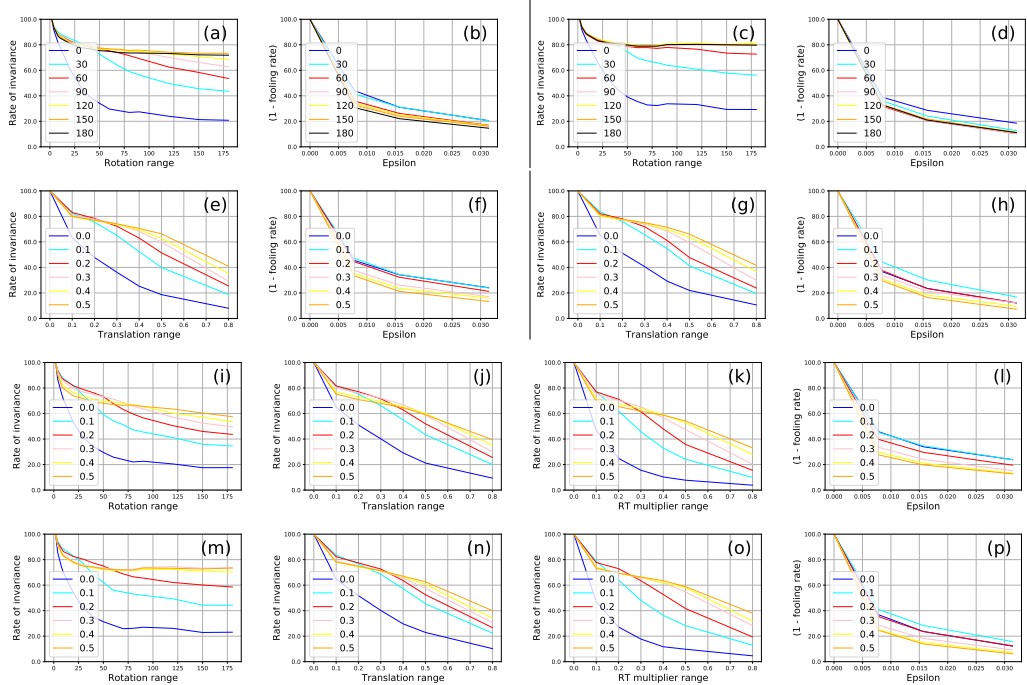

Figure 10: *(Best viewed in color, zoomed in)* On CIFAR100, For VGG16 model (a-b) **Aug - R** StdCNN, (c-d) **Aug - R** GCNN, (e-f) **Aug - T** StdCNN, (g-h) **Aug - T** GCNN, (i-l) **Aug - RT** StdCNN, (m-p) **Aug - RT**, invariance profiles of StdCNN/GCNN models and corresponding robustness profiles.

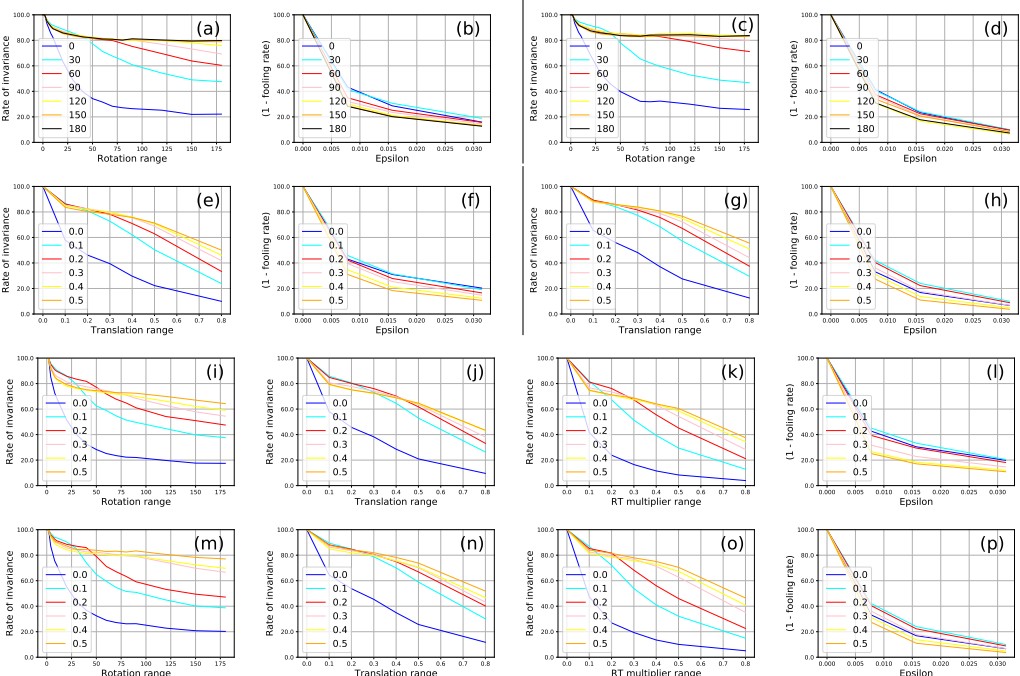

Figure 11: *(Best viewed in color, zoomed in)* On CIFAR100, For ResNet18 model (a-b) **Aug - R** StdCNN, (c-d) **Aug - R** GCNN, (e-f) **Aug - T** StdCNN, (g-h) **Aug - T** GCNN, (i-l) **Aug - RT** StdCNN, (m-p) **Aug - RT**, invariance profiles of StdCNN/GCNN models and corresponding robustness profiles.

vectors obtained by DeepFool [33] for the datapoints under attack. We take the norm of this attack vector as an approximation to the shortest distance of the test point to the decision boundary. For

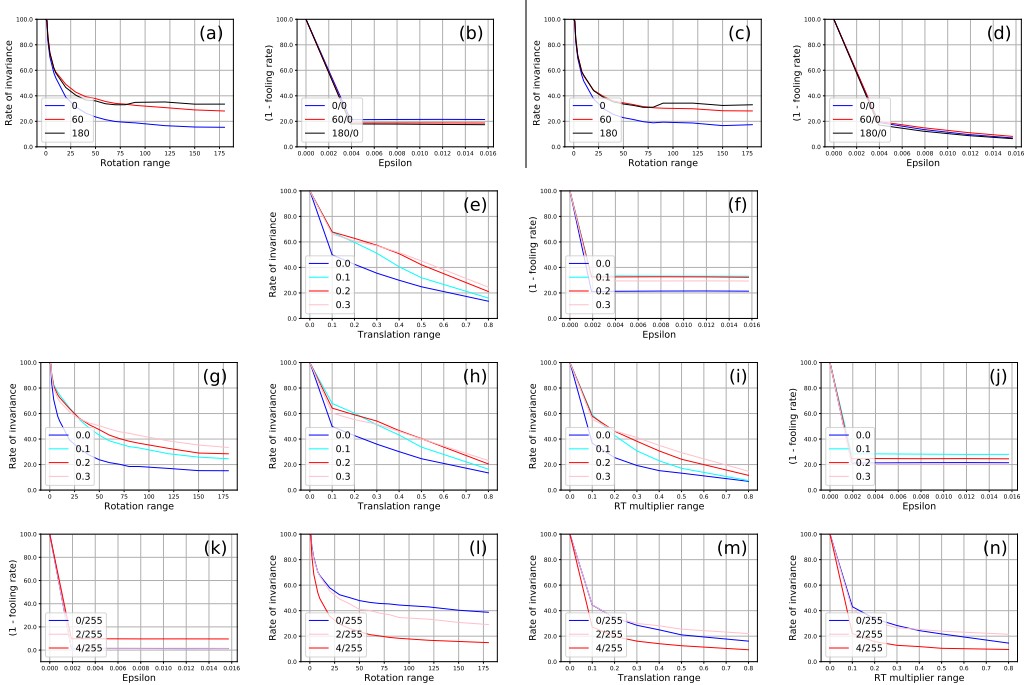

Figure 12: *(Best viewed in color, zoomed in)* On Tiny-ImageNet for ResNet18 model: (a-b) **Aug - R** StdCNN, (c-d) **Aug - R** GCNN, (e-f) **Aug - T** StdCNN, (g-j) **Aug - RT** StdCNN, (k-n) **Adv - PGD** StdCNN, invariance profiles of StdCNN/GCNN models and corresponding robustness profiles.

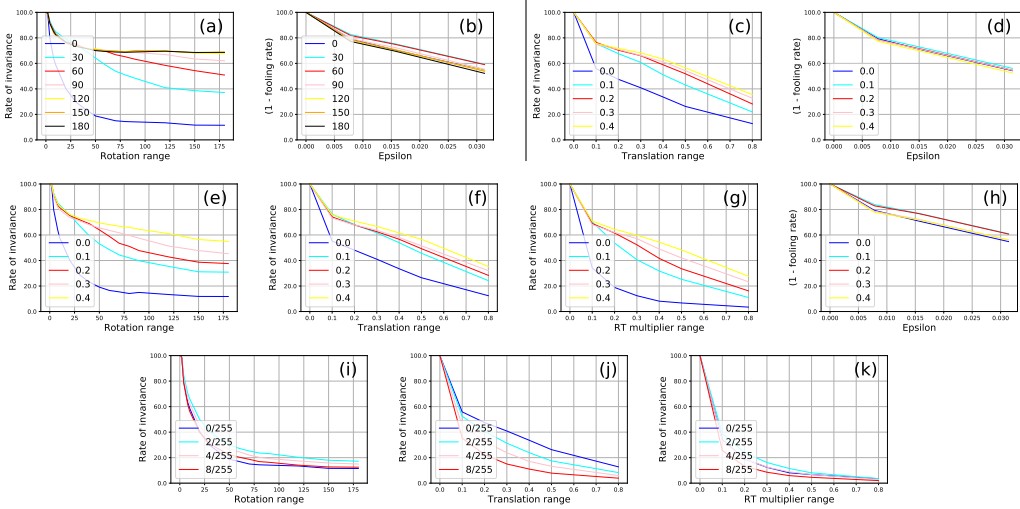

Figure 13: *(Best viewed in color, zoomed in)* On Tiny-ImageNet for Densenet121 model: (a-b) **Aug - R** StdCNN, (c-d) **Aug - T** StdCNN, (e-h) **Aug - RT** StdCNN, (i-k) **Adv - PGD** StdCNN, invariance profiles of StdCNN models and corresponding robustness profiles.

each of the test points we collect the perturbation vectors given by DeepFool attack and report the average perturbation distance. We plot this average distance as the datasets are augmented with larger rotations. Our experiments show that as the networks learn larger rotations with augmentation, the average perturbation distance falls. So as (symmetric) networks become invariant to rotations, they are more vulnerable to pixel-wise attacks. The plots in Fig 16 show this for StdCNNs and GCNNs on MNIST, CIFAR10 and CIFAR100. We plot the accuracy of these models and also the average perturbation distance of the test points alongside in one figure, e.g. 16(a) shows accuracy and 16(b) shows corresponding average perturbation distance. The blue line in Fig 16(a) shows the accuracy of

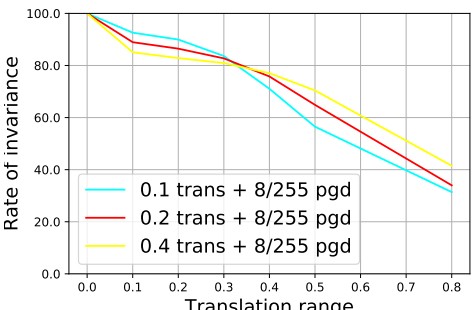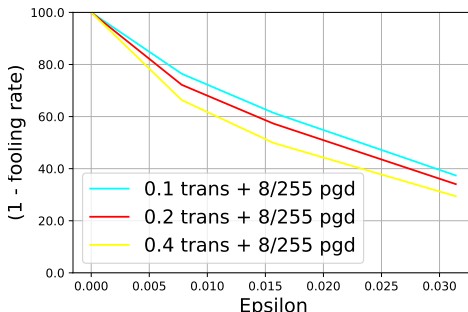

Figure 14: On CIFAR10, **Aug - T + Adv - PGD** ($\epsilon = 8/255$); *(Left)* Translation invariance profiles and *(Right)* robustness profiles of StdCNN/VGG16 models. Different colored lines represent models initially trained with different translation augmentation to make them translation invariant first. Each model made invariant to different translation ranges was further adversarially trained with $\ell_\infty$ budget $\epsilon = 8/255$.

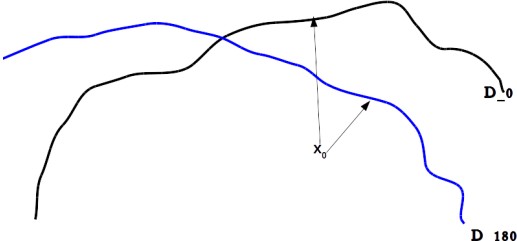

Figure 15: Distance of point $x_0$ to decision boundary $D_{180}$ obtained by augmenting training set with random rotations in range $[-180°, 180°]$ is different compared to the decision boundary $D_0$ obtained with no training augmentation. (This figure is only for illustrative purposes, and does not reflect actual measurements.)

a StdCNN on MNIST when the training data is augmented with $\theta$ ranging from 0 to 180 and the test has no augmentation. The red line shows the accuracy when the test is not augmented with rotations but is PGD perturbed with $\ell_\infty$ norm 0.3. The red line in Fig 16(b) shows the average perturbation distance of the unrotated test points when the network is trained with rotations upto $\theta$ - this is about 5 when $\theta$ is $0°$ (the point on $y$-axis where curves begin). As the network is trained with random rotations up to $180°$, the average perturbation distance of the augmented test drops below 4.0. Fig 16(a) shows that that the PGD accuracy has dropped from around 65% for the network at $0°$ to 20% at $180°$.

## M  Curriculum Learning-based Approach for Pareto-Optimality: Additional Results

In the main paper, we presented results of CuSP based on PGD for the CIFAR10 dataset trained with StdCNN/VGG16 architecture. In this section, we provide additional experimental results and visualizations based on CuSP with PGD and TRADES on StdCNN/ResNet18 for the CIFAR10 and CIFAR100 datasets. We also include additional results using TRADES on StdCNN/VGG16 for the CIFAR10 dataset. Tables 4 and 5 contain the results for StdCNN/VGG16 and StdCNN/ResNet18 on CIFAR10, respectively. (Note that Table 4 is a more detailed version of Table 1 in the main paper). Table 6 presents the results for StdCNN/ResNet18 on CIFAR100. Similar to Fig 6 in the main paper, Figs 17, 18, 19, 20, 21 and 22 present a visualization of the results to understand their relevance to the Pareto frontier.

*Observations:* In Table 4 and 5, the rows indexed 5 and 6 indicate the training strategy of sequentially combining adversarial training and augmentation by spatial transformations (as in Table 1 of the main paper), i.e., take an adversarially trained network and re-train it with spatial (e.g., rotation) augmentation, or vice versa. It is clear from the visualizations in Figs 17, 18, 19 and 20 that they

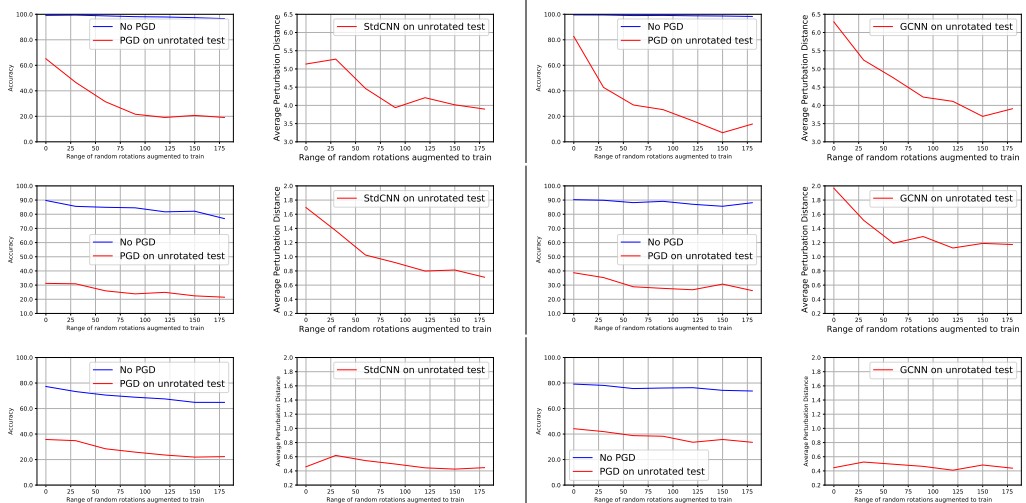

Figure 16: *(Best viewed in color, zoomed in)* Accuracy of StdCNN/GCNN on MNIST(Table2)/CIFAR10(VGG16)/CIFAR100(ResNet18) with/without PGD ($\epsilon = 0.3, \epsilon = 8/255, \epsilon = 2/255$), on unrotated test. Train/test if augmented are with random rotations in $[-\theta°, \theta°]$. (a-d) MNIST, (e-h) CIFAR10, (i-l) CIFAR100, with each pair of plots eg. (a) being the accuracy while (b) being the corresponding avg. perturbation distance.

produce models which lie either bottom-right of the plot (high spatial accuracy but low adversarial robustness) or top-left of the plot (high adversarial robustness but low spatial accuracy). These are examples of the version of catastrophic forgetting displayed by these networks, as discussed earlier. Another viable baseline for our study is adversarial training with data augmented with rotations in the range $[-180°, +180°]$. We observe from these results that CuSP with various configurations produces a model with both spatial invariance and adversarial robustness, in comparison to the baselines. While rows indexed 4 and 6 across the tables seem to show promise, *PGD(Aug 180)* expectedly suffers from relatively poorer adversarial accuracy, while *Aug 180 → PGD (Aug 0)* expectedly suffers from poor spatial accuracy. This is clearly a case of the aforementioned catastrophic forgetting issue, as the latter has a behavior similar to row indexed 3 *PGD(Aug 0)*. The proposed method, CuSP, consistently outperforms these methods across the datasets and models – supporting the role of curriculum learning for simultaneous robustness on multiple fronts.

*Other Settings and Extensions:* While the main set of experiments were based on TRADES learning rate schedule 75-90-100, we also tried other schedules like 40-80 (40-th and 80-th epochs) and 50-100. Table 7 shows that these also obtain competent results, hence indicating that CuSP can be further finetuned to obtain better Pareto-optimal solutions, which is a promising direction of future work.

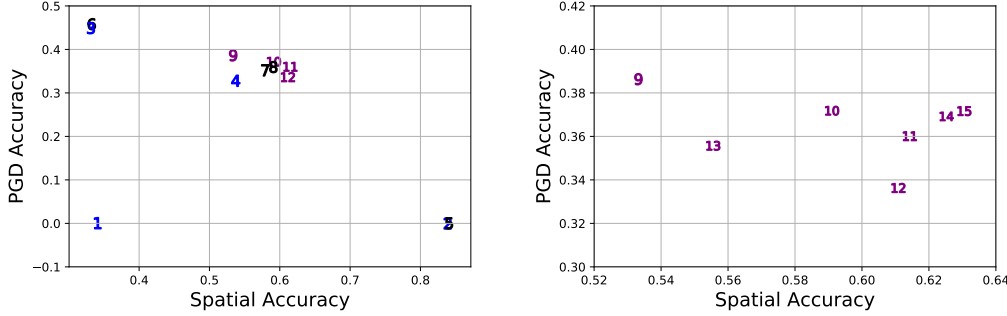

Figure 17: Visualization of performance of CuSP based on PGD against other baseline strategies on CIFAR10 for StdCNN/VGG16 model (each index corresponds to a row in Table 4). On the left, we compare CuSP against various baselines. On the right, we zoom in to compare different variants of CuSP.

| | Training Method | Adv (PGD) Accuracy(%) | Std Accuracy(%) | Spatial Accuracy(%) |
|---|---|---|---|---|
| 1 | Natural (Aug 0) | 00.05 | 92.89 | 33.99 |
| 2 | Natural (Aug 180) | 00.00 | 85.00 | 83.78 |
| 3 | PGD (Aug 0) | 44.88 | 80.08 | 33.07 |
| 4 | PGD (Aug 180) | 32.79 | 54.60 | 53.69 |
| 5 | PGD (Aug 0) $\to$ Aug 180 | 00.00 | 85.46 | 83.99 |
| 6 | Aug 180 $\to$ PGD (Aug 0) | 45.80 | 81.03 | 33.19 |
| 7 | PGD (Aug 0) $\to$ PGD (Aug 180) | 35.19 | 58.96 | 57.84 |
| 8 | Aug 180 $\to$ PGD (Aug 180) | 35.94 | 60.27 | 59.02 |
| 9 | CuSP (PGD, 30-60-180, $\{\frac{2}{255}, \frac{4}{255}, \frac{8}{255}\}$) | 38.63 | 65.92 | 53.31 |
| 10 | CuSP (PGD, 60-120-180, $\{\frac{2}{255}, \frac{4}{255}, \frac{8}{255}\}$) | 37.18 | 65.88 | 59.09 |
| 11 | CuSP (PGD, 120-150-180, $\{\frac{2}{255}, \frac{4}{255}, \frac{8}{255}\}$) | 36.01 | 64.96 | 61.41 |
| 12 | CuSP (PGD, 180, $\{\frac{2}{255}, \frac{4}{255}, \frac{8}{255}\}$) | 33.63 | 62.17 | 61.07 |
| 13 | CuSP (PGD, 120-150-180, $\{\frac{8}{255}\}$) | 35.57 | 58.08 | 55.54 |
| 14 | PGD (Aug 0) $\to$ CuSP (PGD, 120-150-180, $\{\frac{2}{255}, \frac{4}{255}, \frac{8}{255}\}$) | 36.92 | 66.25 | 62.51 |
| 15 | Aug 180 $\to$ Aug 180 $\to$ CuSP (PGD, 120-150-180, $\{\frac{2}{255}, \frac{4}{255}, \frac{8}{255}\}$) | 37.16 | 66.58 | 63.04 |
| 1 | Natural (Aug 0) | 00.05 | 92.89 | 33.99 |
| 2 | Natural (Aug 180) | 00.00 | 85.00 | 83.78 |
| 3 | TRADES (Aug 0) | 46.90 | 79.35 | 32.20 |
| 4 | TRADES (Aug 180) | 29.93 | 62.38 | 61.85 |
| 5 | TRADES (Aug 0) $\to$ Aug 180 | 00.00 | 87.82 | 86.22 |
| 6 | Aug 180 $\to$ TRADES (Aug 0) | 47.61 | 79.90 | 32.65 |
| 7 | TRADES (Aug 0) $\to$ TRADES (Aug 180) | 31.43 | 63.77 | 63.70 |
| 8 | Aug 180 $\to$ TRADES (Aug 180) | 30.65 | 63.37 | 63.38 |
| 9 | CuSP (TRADES, 30-60-180, $\{\frac{2}{255}, \frac{4}{255}, \frac{8}{255}\}$) | 35.69 | 71.47 | 59.81 |
| 10 | CuSP (TRADES, 60-120-180, $\{\frac{2}{255}, \frac{4}{255}, \frac{8}{255}\}$) | 32.76 | 69.80 | 63.62 |
| 11 | CuSP (TRADES, 120-150-180, $\{\frac{2}{255}, \frac{4}{255}, \frac{8}{255}\}$) | 31.36 | 68.04 | 65.53 |
| 12 | CuSP (TRADES, 180, $\{\frac{2}{255}, \frac{4}{255}, \frac{8}{255}\}$) | 29.31 | 66.14 | 65.29 |
| 13 | CuSP (TRADES, 120-150-180, $\{\frac{8}{255}\}$) | 31.82 | 65.17 | 62.87 |
| 14 | TRADES (Aug 0) $\to$ CuSP (TRADES, 120-150-180, $\{\frac{2}{255}, \frac{4}{255}, \frac{8}{255}\}$) | 33.61 | 69.56 | 66.84 |
| 15 | Aug 180 $\to$ Aug 180 $\to$ CuSP (TRADES, 120-150-180, $\{\frac{2}{255}, \frac{4}{255}, \frac{8}{255}\}$) | 32.10 | 69.29 | 66.00 |

Table 4: Comparison of performance of proposed CuSP using StdCNN/VGG16 with other baseline strategies on CIFAR10. (Aug $\theta$): denotes training data augmented with random rotations in the range $[-\theta, +\theta]$; $a \to b$: denotes $a$ sequentially followed by $b$ during training.

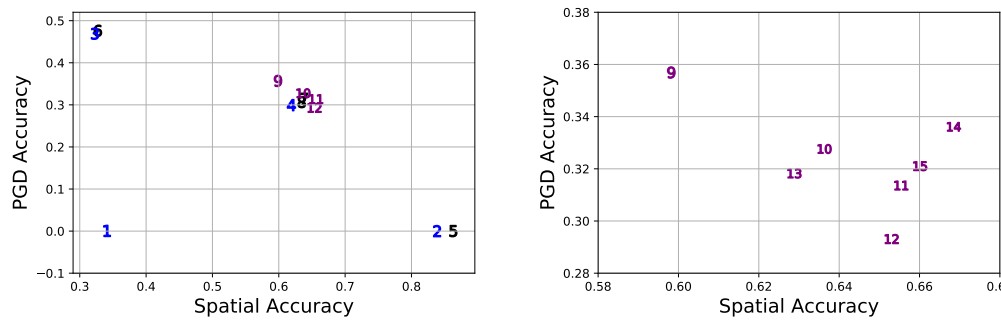

Figure 18: Visualization of performance of CuSP based on TRADES against other baseline strategies on CIFAR10 for StdCNN/VGG16 model (each index corresponds to a row in Table 4). On the left, we compare CuSP against various baselines. On the right, we zoom in to compare different variants of CuSP.

| | Training Method | Adv (PGD) Accuracy(%) | Std Accuracy(%) | Spatial Accuracy(%) |
|---|---|---|---|---|
| 1 | Natural (Aug 0) | 00.00 | 94.44 | 35.35 |
| 2 | Natural (Aug 180) | 00.00 | 86.99 | 85.65 |
| 3 | PGD (Aug 0) | 47.67 | 83.72 | 33.50 |
| 4 | PGD (Aug 180) | 38.00 | 65.30 | 64.53 |
| 5 | PGD (Aug 0) $\rightarrow$ Aug 180 | 00.00 | 87.90 | 86.80 |
| 6 | Aug 180 $\rightarrow$ PGD (Aug 0) | 47.62 | 83.73 | 34.63 |
| 7 | PGD (Aug 0) $\rightarrow$ PGD (Aug 180) | 38.70 | 65.88 | 64.94 |
| 8 | Aug 180 $\rightarrow$ PGD (Aug 180) | 38.62 | 67.79 | 67.10 |
| 9 | CuSP (PGD, 30-60-180, $\{\frac{2}{255}, \frac{4}{255}, \frac{8}{255}\}$) | 41.05 | 71.66 | 59.96 |
| 10 | CuSP (PGD, 60-120-180, $\{\frac{2}{255}, \frac{4}{255}, \frac{8}{255}\}$) | 40.12 | 71.57 | 66.13 |
| 11 | CuSP (PGD, 120-150-180, $\{\frac{2}{255}, \frac{4}{255}, \frac{8}{255}\}$) | 38.84 | 71.31 | 68.67 |
| 12 | CuSP (PGD, 180, $\{\frac{2}{255}, \frac{4}{255}, \frac{8}{255}\}$) | 36.17 | 69.65 | 68.85 |
| 13 | CuSP (PGD, 120-150-180,$\{\frac{8}{255}\}$) | 40.36 | 67.32 | 65.07 |
| 14 | PGD (Aug 0) $\rightarrow$ PGD (Aug 0) $\rightarrow$ CuSP (PGD, 120-150-180, $\{\frac{2}{255}, \frac{4}{255}, \frac{8}{255}\}$) | 38.92 | 70.65 | 68.07 |
| 15 | Aug 180 $\rightarrow$ CuSP (PGD, 120-150-180, $\{\frac{2}{255}, \frac{4}{255}, \frac{8}{255}\}$) | 38.48 | 72.27 | 70.15 |
| 1 | Natural (Aug 0) | 00.00 | 94.44 | 35.35 |
| 2 | Natural (Aug 180) | 00.00 | 86.99 | 85.65 |
| 3 | TRADES (Aug 0) | 50.92 | 81.83 | 32.96 |
| 4 | TRADES (Aug 180) | 36.13 | 68.15 | 68.22 |
| 5 | TRADES (Aug 0) $\rightarrow$ Aug 180 | 00.00 | 89.46 | 88.33 |
| 6 | Aug 180 $\rightarrow$ TRADES (Aug 0) | 51.55 | 82.39 | 33.75 |
| 7 | TRADES (Aug 0) $\rightarrow$ TRADES (Aug 180) | 34.04 | 66.13 | 66.29 |
| 8 | Aug 180 $\rightarrow$ TRADES (Aug 180) | 34.11 | 65.67 | 65.48 |
| 9 | CuSP (TRADES, 30-60-180, $\{\frac{2}{255}, \frac{4}{255}, \frac{8}{255}\}$) | 39.92 | 73.79 | 63.17 |
| 10 | CuSP (TRADES, 60-120-180, $\{\frac{2}{255}, \frac{4}{255}, \frac{8}{255}\}$) | 38.41 | 73.16 | 67.46 |
| 11 | CuSP (TRADES, 120-150-180, $\{\frac{2}{255}, \frac{4}{255}, \frac{8}{255}\}$) | 36.63 | 71.76 | 69.10 |
| 12 | CuSP (TRADES, 180, $\{\frac{2}{255}, \frac{4}{255}, \frac{8}{255}\}$) | 34.53 | 70.19 | 69.66 |
| 13 | CuSP (TRADES, 120-150-180,$\{\frac{8}{255}\}$) | 38.10 | 69.97 | 67.77 |
| 14 | TRADES (Aug 0) $\rightarrow$ CuSP (TRADES, 120-150-180, $\{\frac{2}{255}, \frac{4}{255}, \frac{8}{255}\}$) | 35.33 | 70.64 | 68.44 |
| 15 | Aug 180 $\rightarrow$ CuSP (TRADES, 120-150-180, $\{\frac{2}{255}, \frac{4}{255}, \frac{8}{255}\}$) | 35.99 | 71.37 | 69.34 |

Table 5: Comparison of performance of proposed CuSP using StdCNN/ResNet18 with other baseline strategies on CIFAR10. (Aug $\theta$): denotes training data augmented with random rotations in the range $[-\theta, +\theta]$; $a \rightarrow b$: denotes $a$ sequentially followed by $b$ during training.

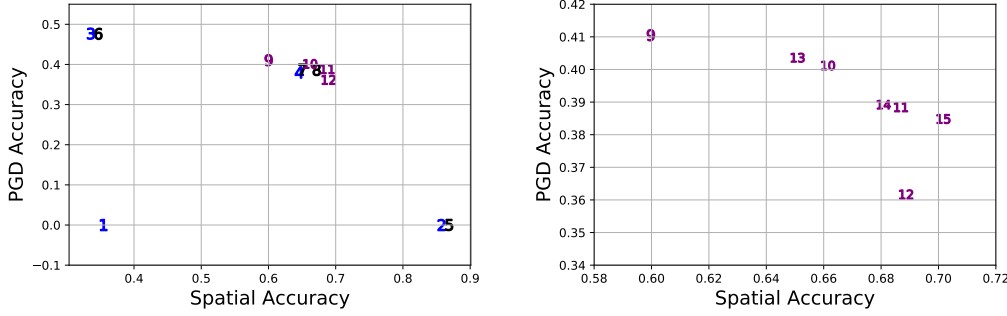

Figure 19: Visualization of performance of CuSP based on PGD against other baseline strategies on CIFAR10 for StdCNN/ResNet18 model (each index corresponds to a row in Table 5). On the left, we compare CuSP against various baselines. On the right, we zoom in to compare different variants of CuSP.

| | Training Method | Adv (PGD) Accuracy(%) | Std Accuracy(%) | Spatial Accuracy(%) |
|---|---|---|---|---|
| 1 | Natural (Aug 0) | 00.06 | 74.93 | 20.27 |
| 2 | Natural (Aug 180) | 00.01 | 58.96 | 59.49 |
| 3 | PGD (Aug 0) | 23.28 | 56.27 | 19.44 |
| 4 | PGD (Aug 180) | 15.96 | 41.48 | 41.39 |
| 5 | PGD (Aug 0) $\rightarrow$ Aug 180 | 00.02 | 64.24 | 64.17 |
| 6 | Aug 180 $\rightarrow$ PGD (Aug 0) | 22.79 | 56.48 | 20.24 |
| 7 | PGD (Aug 0) $\rightarrow$ PGD (Aug 180) | 20.03 | 44.24 | 44.19 |
| 8 | Aug 180 $\rightarrow$ PGD (Aug 180) | 19.04 | 43.82 | 43.79 |
| 9 | CuSP (PGD, 30-60-180, $\{\frac{2}{255}, \frac{4}{255}, \frac{8}{255}\}$) | 19.97 | 51.35 | 42.31 |
| 10 | CuSP (PGD, 120-150-180, $\{\frac{2}{255}, \frac{4}{255}, \frac{8}{255}\}$) | 17.13 | 49.80 | 48.64 |
| 11 | CuSP (PGD, 180, $\{\frac{2}{255}, \frac{4}{255}, \frac{8}{255}\}$) | 15.86 | 48.34 | 48.14 |
| 12 | CuSP (PGD, 120-150-180, $\{\frac{8}{255}\}$) | 20.69 | 43.95 | 43.06 |
| 13 | PGD (Aug 0) $\rightarrow$ CuSP (PGD, 120-150-180, $\{\frac{2}{255}, \frac{4}{255}, \frac{8}{255}\}$) | 17.88 | 50.10 | 49.15 |
| 14 | Aug 180 $\rightarrow$ CuSP (PGD, 120-150-180, $\{\frac{2}{255}, \frac{4}{255}, \frac{8}{255}\}$) | 15.80 | 49.85 | 49.17 |
| 1 | Natural (Aug 0) | 00.06 | 74.93 | 20.27 |
| 2 | Natural (Aug 180) | 00.01 | 58.96 | 59.49 |
| 3 | TRADES (Aug 0) | 27.12 | 55.55 | 19.05 |
| 4 | TRADES (Aug 180) | 18.98 | 44.75 | 44.82 |
| 5 | TRADES (Aug 0) $\rightarrow$ Aug 180 | 00.01 | 64.84 | 64.13 |
| 6 | Aug 180 $\rightarrow$ TRADES (Aug 0) | 27.01 | 55.10 | 19.83 |
| 7 | TRADES (Aug 0) $\rightarrow$ TRADES (Aug 180) | 19.27 | 46.46 | 46.08 |
| 8 | Aug 180 $\rightarrow$ TRADES (Aug 180) | 18.69 | 45.74 | 46.13 |
| 9 | CuSP (TRADES, 30-60-180, $\{\frac{2}{255}, \frac{4}{255}, \frac{8}{255}\}$) | 20.00 | 50.43 | 42.17 |
| 10 | CuSP (TRADES, 120-150-180, $\{\frac{2}{255}, \frac{4}{255}, \frac{8}{255}\}$) | 18.00 | 48.83 | 47.34 |
| 11 | CuSP (TRADES, 180, $\{\frac{2}{255}, \frac{4}{255}, \frac{8}{255}\}$) | 16.57 | 46.69 | 47.06 |
| 12 | CuSP (TRADES, 120-150-180, $\{\frac{8}{255}\}$) | 20.09 | 46.61 | 45.63 |
| 13 | TRADES (Aug 0) $\rightarrow$ CuSP (TRADES, 120-150-180, $\{\frac{2}{255}, \frac{4}{255}, \frac{8}{255}\}$) | 18.57 | 48.90 | 47.60 |
| 14 | Aug 180 $\rightarrow$ CuSP (TRADES, 120-150-180, $\{\frac{2}{255}, \frac{4}{255}, \frac{8}{255}\}$) | 17.64 | 48.97 | 47.40 |

Table 6: Comparison of performance of proposed CuSP using StdCNN/ResNet18 with other baseline strategies on CIFAR100. (Aug $\theta$): denotes training data augmented with random rotations in the range $[-\theta, +\theta]$; $a \rightarrow b$: denotes $a$ sequentially followed by $b$ during training.

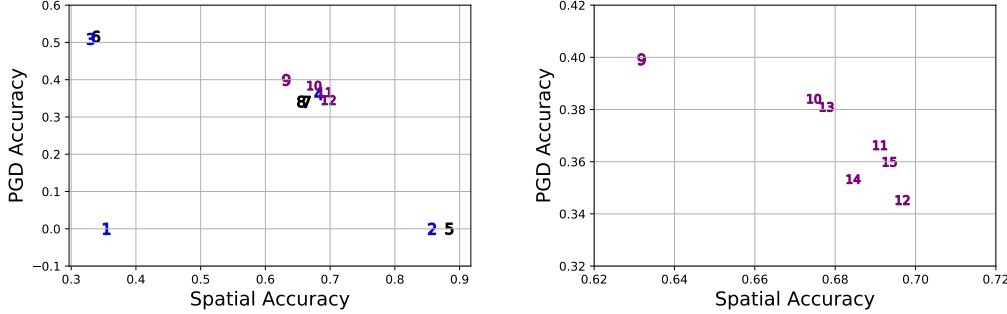

Figure 20: Visualization of performance of CuSP based on TRADES against other baseline strategies on CIFAR10 for StdCNN/ResNet18 model (each index corresponds to a row in Table 5). On the left, we compare CuSP against various baselines. On the right, we zoom in to compare different variants of CuSP.

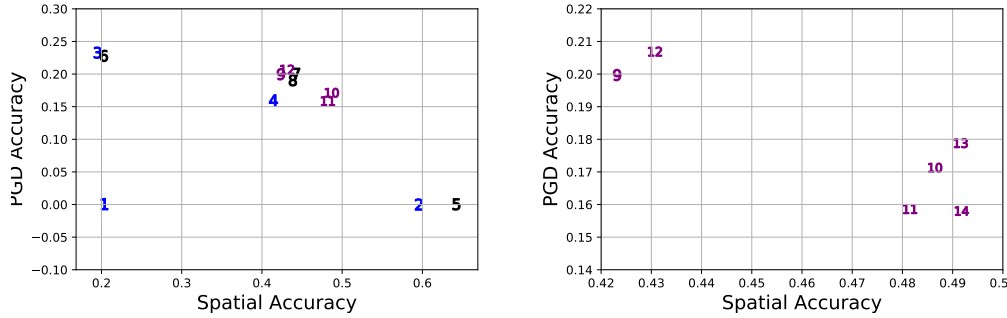

Figure 21: Visualization of performance of CuSP based on PGD against other baseline strategies on CIFAR100 for StdCNN/ResNet18 model (each index corresponds to a row in Table 6). On the left, we compare CuSP against various baselines. On the right, we zoom in to compare different variants of CuSP.

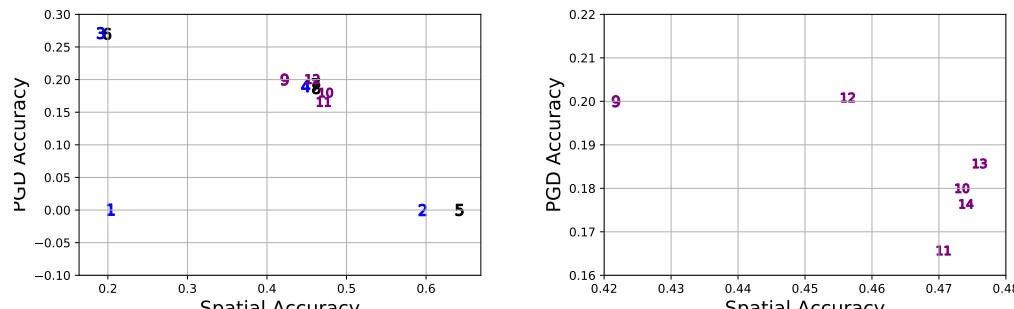

Figure 22: Visualization of performance of CuSP based on TRADES against other baseline strategies on CIFAR100 for StdCNN/ResNet18 model (each index corresponds to a row in Table 6). On the left, we compare CuSP against various baselines. On the right, we zoom in to compare different variants of CuSP.

| Learning Rate Schedule | Training Method | Adv (PGD) Accuracy(%) | Std Accuracy(%) | Spatial Accuracy(%) |
|---|---|---|---|---|
| $[75 - 90 - 100]$ | PGD (Aug 180) | 32.79 | 54.60 | 53.69 |
| $[50 - 100]$ | PGD (Aug 180) | 34.17 | 57.48 | 56.97 |
| $[40 - 80]$ | PGD (Aug 180) | 34.00 | 57.40 | 56.40 |
| $[75 - 90 - 100]$ | CuSP (PGD, 180, $\{\frac{2}{255}, \frac{4}{255}, \frac{8}{255}\}$) | 33.63 | 62.17 | 61.07 |
| $[50 - 100]$ | CuSP (PGD, 180, $\{\frac{2}{255}, \frac{4}{255}, \frac{8}{255}\}$) | 34.86 | 65.09 | 63.87 |
| $[40 - 80]$ | CuSP (PGD, 180, $\{\frac{2}{255}, \frac{4}{255}, \frac{8}{255}\}$) | 34.06 | 61.97 | 61.21 |

Table 7: Performance of proposed CuSP vs PGD (Aug 180) for CIFAR10 on StdCNN/VGG16 with various learning rate schedules (Aug $\theta$): denotes training data augmented with random rotations in the range $[-\theta, +\theta]$; $a \rightarrow b$: denotes $a$ sequentially followed by $b$ during training; $[E_1 - E_2 - ... - E_h]$ : denotes the learning rate changes at epoch $E_i$.

| Network Architecture | Training Method !h | Adv (PGD) Accuracy(%) | Std Accuracy(%) | Spatial Accuracy(%) |
|---|---|---|---|---|
| LeNet (6-16-120-84-10) | PGD(Aug 0) | 27.38 | 40.52 | 23.36 |
| LeNet (6-16-120-84-10) | PGD(Aug 180) | 20.80 | 26.94 | 27.18 |
| MLP (2048-2048-240-168-10) | PGD(Aug 0) | 28.76 | 44.64 | 24.66 |
| MLP (2048-2048-240-168-10) | PGD(Aug 180) | 19.51 | 25.66 | 25.89 |
| MLP (4096-4096-480-336-10) | PGD(Aug 0) | 28.89 | 45.34 | 25.31 |
| MLP (4096-4096-480-336-10) | PGD(Aug 180) | 19.87 | 26.77 | 26.75 |

Table 8: Trade-off (PGD(Aug 0) vs PGD(Aug 180)) on simpler architectures like LeNet and MLP on CIFAR10 dataset.

# N   Additional Experiments on Different Architectures

To study the trade-off further across a wider variety of architectures, we ran additional experiments using LeNet (2 Conv + 2 FullyConnected) and two MLP (4 layers) architectures and obtain the following results in Table 8 similar to Table 1 row 3 (PGD (Aug 0)) and row 4 (PGD (Aug 180)) with CIFAR10. Note that the trade-off exists here too, supporting our observations across a gradation of architectures (MLP, LeNet, VGG16, ResNet18, WideResNet34, GCNN).