# OpenReview forum: "Can we have it all? On the Trade-off between Spatial and Adversarial Robustness of Neural Networks"
_NeurIPS.cc/2021/Conference — NeurIPS 2021 Poster_

### Official Review · Reviewer_YXLX · 2021-07-10

**Rating:** 5
**Confidence:** 4

**Summary:**

The authors study the trade-off between spatial (translation and rotation) robustness and adversarial robustness. They show theoretical proofs and empirical evidence that two properties may be at odds with each other. A curriculum lraining based method is proposed to mitigate the problem.

**Limitations And Societal Impact:**

The authors discuss the societal impact. I didn't see any section / paragraph of the discussion on the limitations of the work.

**Main Review:**

[Theoretical results]

The theoretical results are interesting but not substantial. The authors study the bayes optimal classifier in a simple setting and show that spatial robustness is non-trivial to achieve and high adversarial accuracy leads to low spatial accuracy. However, the low spatial accuracy is mostly due to the decrease in the standard accuracy. To see this, since the authors consider the average-case spatial robustness which depends on the accuracy of both translated and original data, the accuracy on the original data play a larger role when the number of possible translations is smaller, i.e. $m$ is larger, which is exactly when Theorem 2 supports the claim. This also can be seen in the proof (line 566) where the authors bound the accuracy of the translated data to be 1. In general, the setting with $m=1$ might be more practical and hence interesting as it considers all the possible translations.

[Experiments]

The authors train stdCNNs and GCNNs with different levels of data augmentation to achieve different levels of spatial / adversarial robustness. The results seem to support the claim on the trade off between two robustness but only in a sense of correlation rather than causality. For example, the observation that the larger translation robustness and worse adversarial robustness can be both attributed to the stronger data augmentation [1, 2]. A more convincing proof needs more carefully controlled factors, for example, the authors may consider models with inherently different levels of spatial robustness like fully connected networks and ViTs [3].

Sometimes slightly improving the spatial robustness improves the adversarial robustness, e.g. Fig. 7 (f, h, p), which seems to be at odd with the claim.

Minor - Figure 7 caption: "(m-p) Aug - RT" --> "(m-p) Aug - RT GCNN".

[Originality]
The proposed curriculum lraining based method is novel and interesting. There is an ongoing discussion on the trade-off between spatial and adversarial robustness and it might be interesting to discuss the relationship to these works [4, 5].

[1] Chen, Lin, et al. "More data can expand the generalization gap between adversarially robust and standard models." International Conference on Machine Learning. PMLR, 2020.

[2] Rebuffi, Sylvestre-Alvise, et al. "Fixing data augmentation to improve adversarial robustness." arXiv preprint arXiv:2103.01946 (2021).

[3] Shao, Rulin, et al. "On the adversarial robustness of visual transformers." arXiv preprint arXiv:2103.15670 (2021).Shao, Rulin, et al. "On the adversarial robustness of visual transformers." arXiv preprint arXiv:2103.15670 (2021).

[4] Ge, Songwei, et al. "Shift Invariance Can Reduce Adversarial Robustness." arXiv preprint arXiv:2103.02695 (2021).

[5] Kamath, Sandesh, Amit Deshpande, and K. V. Subrahmanyam. "Invariance vs Robustness of Neural Networks." (2019).


**Time Spent Reviewing:**

6

---

> ### Author Response · Authors · 2021-08-10
> **Response to Reviewer YXLX**
>
> >[Theoretical results]
> The theoretical results are interesting but not substantial. The authors study the bayes optimal classifier in a simple setting and show that spatial robustness is non-trivial to achieve and high adversarial accuracy leads to low spatial accuracy. However, the low spatial accuracy is mostly due to the decrease in the standard accuracy. To see this, since the authors consider the average-case spatial robustness which depends on the accuracy of both translated and original data, the accuracy on the original data play a larger role when the number of possible translations is smaller, i.e., $m$
> is larger, which is exactly when Theorem 2 supports the claim. This also can be seen in the proof (line 566) where the authors bound the accuracy of the translated data to be 1. In general, the setting with m=1 might be more practical and hence interesting as it considers all the possible translations.
>
>
>
> Our theoretical results are novel, and work for the common transformation groups used in the theory of group-equivariant CNNs, e.g., $m=d/4$ corresponds to the *p*4 group of rotations that are integer multiples of $90^{\circ}$. We agree that the case of small $m$  more closely approximates true invariance to all rotations (or translations), and we consider it as future work. (We note that although weaker than ideal, the results in our work are non-trivial and have not been shown before.)
>
>
> >[Experiments]
> The authors train stdCNNs and GCNNs with different levels of data augmentation to achieve different levels of spatial / adversarial robustness. The results seem to support the claim on the trade off between two robustness but only in a sense of correlation rather than causality. For example, the observation that the larger translation robustness and worse adversarial robustness can be both attributed to the stronger data augmentation [1, 2]. A more convincing proof needs more carefully controlled factors, for example, the authors may consider models with inherently different levels of spatial robustness like fully connected networks and ViTs [3].
>
> We show that adversarial training with gradually increasing perturbation budget gives gradually better adversarial robustness but gradually worse spatial robustness (see Figs 2, 3, 4). This side of the trade-off is independent of any data augmentation. On the other hand, better spatial robustness requires larger data augmentation (in addition to better or equivariant models). For better spatial robustness, one may not be able to easily decouple the effect of model (e.g., fully connected networks, StdCNN, GCNN) from that of data augmentation. We however show empirically that our observations hold consistently across models popularly used in practice (VGG16, ResNet18, WideResNet34, GCNN) across multiple datasets (MNIST, CIFAR10, CIFAR100, Tiny-Imagenet).
>
> To study this further across a wider variety of architectures, we ran additional experiments using LeNet (2 Conv + 2 FullyConnected) and two MLP (4 layers) architectures similar to Table 1 row 1 (PGD (Aug 0)) and row 2 (PGD (Aug 180)) with CIFAR10 and obtained the following results. Note that the trade-off exists here too, supporting our observations across a gradation of architectures (MLP, LeNet, VGG16, ResNet18, WideResNet34, GCNN).
>
> |Net Arch | Training Method | Adv (PGD) Accuracy(%) | Std Accuracy(%) | Spatial Accuracy(%) |
> |-------|---------------|-------|-------|-------|
> | LeNet (6-16-120-84-10)      | PGD(Aug 0)   | 27.38 | 40.52 | 23.36 |
> | LeNet (6-16-120-84-10)      | PGD(Aug 180) | 20.80 | 26.94 | 27.18 |
> | MLP (2048-2048-240-168-10)  | PGD(Aug 0)   | 28.76 | 44.64 | 24.66 |
> | MLP (2048-2048-240-168-10)  | PGD(Aug 180) | 19.51 | 25.66 | 25.89 |
> | MLP (4096-4096-480-336-10)  | PGD(Aug 0)   | 28.89 | 45.34 | 25.31 |
> | MLP (4096-4096-480-336-10)  | PGD(Aug 180) | 19.87 | 26.77 | 26.75 |
>
> Our theoretical analysis holds for cyclic groups of transformations (e.g., rotations by multiples of $90^{\circ}$) used in the theory of GCNNs (ref. [8]), so StdCNNs and GCNNs were the natural model choices for our experiments, which we studied with different architectures. Going forward, we will definitely consider ViTs, if our theory can be extended to study invariance to permutations of sequences and other novel notions of invariance.
>
> >Sometimes slightly improving the spatial robustness improves the adversarial robustness, e.g. Fig. 7 (f, h, p), which seems to be at odd with the claim.
>
> Good observation. We too find it intriguing that this happens only in a few cases involving spatial robustness to small translations. Our guess is that when nearby pixel values are similar, spatial robustness to small translations has certain similarities with adversarial robustness to small perturbations of pixel values. However, our overall claim or trends for larger rotations and translations still hold.
>
> >Minor - Figure 7 caption: "(m-p) Aug - RT" --> "(m-p) Aug - RT GCNN".
>
> Thank you for the careful reading. We will update the captions appropriately in the appendix for the final version.
>
> >[Originality] The proposed curriculum lraining based method is novel and interesting. There is an ongoing discussion on the trade-off between spatial and adversarial robustness and it might be interesting to discuss the relationship to these works [4, 5].
>
> >[4] Ge, Songwei, et al. "Shift Invariance Can Reduce Adversarial Robustness." arXiv preprint arXiv:2103.02695 (2021).
>
> >[5] Kamath, Sandesh, Amit Deshpande, and K. V. Subrahmanyam. "Invariance vs Robustness of Neural Networks." (2019).
>
> Thank you for finding CuSP interesting. Our theoretical analysis of the trade-off between spatial and adversarial robustness using cyclic codes and the curriculum learning method we propose are both novel and different from [4,5]. Theoretical analysis in [4] studies the margin between classes for shift-invariant *linear* classifiers. We will cite [4,5] and mention the key differences.

---

> > ### Comment · Reviewer_YXLX · 2021-08-27
> > **Thank you for your response but the justification on the theoretical results is not convincing**
> >
> > I thank the authors for their effort on the responses and additional experiments. However, I am still not convinced that Theorem 2 endorses their claim that high adversarial robustness induces low spatial robustness. If one checks the proof on the lines 566-567 carefully, then it can be seen that the main claim should be that high **adversarial robustness induces low standard accuracy, which is something already known**. For example, considering p4 group of rotations $m=d/4$ and the most favorable conditions that $p\rightarrow 1/2$ and $\eta\rightarrow 0$, the expected accuracy drops to 0 on $0^{\circ}$ rotation (clean example) and stays as 1 on the other rotations, namely ${90^{\circ}, 180^{\circ}, 270^{\circ}}$. This makes the entire spatial accuracy drop to $3/4$. Similar reason causes spatial accuracy on p2 group of rotations to drop to near random. Given that Theorem 2 is the main theoretical result of the paper that supports their claim, I couldn't increase my score more. If there is anything that I misunderstood, please definitely point it out and I'm happy with more discussion.

---

> > > ### Author Response · Authors · 2021-08-27
> > > **Response to Reviewer YXLX**
> > >
> > > Thank you for your response.
> > >
> > > In the observation on lines 566-567 that the accuracy on the clean examples drops to 0 while the accuracy on 90, 180, 270 degree rotations remains 1, please note that these are upper bounds on the respective accuracies. Proposition 1 already shows that there is no model that has accuracy 1 simultaneously on 90, 180, 270 degree rotations. So, the conclusion that "the spatial accuracy drops to $3/4$" is not correct. If it would be useful to explicitly clarify this, we will be happy to do so in the final version. We understand that our upper bounds can possibly be tightened further, but the result has value as is too. We, respectfully, do not agree that Theorem 2 only restates a previously known trade-off between adversarial robustness and standard accuracy.
> > >
> > > Besides, Theorem 2 is not the only justification for our claim. Our extensive experiments, including the ones suggested by the reviewers, show that our claim holds across different models and adversarial attacks. In fact, our additional experiments in response to the original review herein also show that our trade-off is different from the known adversarial robustness and standard accuracy trade-off. The comparison between PGD (Aug 0) and PGD (Aug 180) shows that both adversarial and standard accuracy decrease while spatial accuracy improves.

---

> > > > ### Comment · Reviewer_YXLX · 2021-08-27
> > > > **Response**
> > > >
> > > > Thank you for your answer again.
> > > >
> > > > > please note that these are upper bounds on the respective accuracies.
> > > >
> > > > Yes, I think what we have discussed is all about upper bounds? I don't think there are any proofs of Theorem 2 on the accuracy directly, isn't it?
> > > >
> > > > In other words, if one only considers the accuracy on the spatially transformed data (i.e. excluding clean data), this upper bound will be trivial. Could the authors point out what additional information it brings other than the drops on the upper bound of clean accuracy?
> > > >
> > > > In general, I appreciate the extensive experiments. For this reason, I would like to raise my rating. I also agree that it is useful to have simple theoretical results to help understand the phenomenon. However, I found that the theorem is misleading as its proof does not support the claim that high adversarial robustness induces the accuracy drops on spatially transformed. And the responses of the authors did not address my concern. For this reason, I'm only willing to raise my rating to 5.

---

> > > > > ### Author Response · Authors · 2021-09-01
> > > > > **Response to Reviewer YXLX**
> > > > >
> > > > > The additional information our distribution brings is that it allows simultaneous adversarial robustness vs. accuracy trade-offs for multiple distributions (e.g., 0, 90, 180, 270 degree rotations of the original distribution, resp.) that are sufficiently different from one another (i.e., accuracy close to 1 on any one of them implies a lower accuracy on the others). We use this in the proofs of Proposition 1 and Theorem 2. We agree with your observation that any improvement in our upper bounds (lines 566-567) to closely match the actual accuracies would make Theorem 2 stronger and closer to our experimental observations. (In general, matching theory and experiments with deep neural network models has been a challenging task for researchers in the field.)
> > > > >
> > > > > Please note that the previously known adversarial robustness vs. accuracy trade-offs in theory are for a fixed adversarial perturbation bound $\epsilon$. In our experiments, the adversarial vs. spatial robustness trade-off holds for the entire adversarial robustness profile over varying $\epsilon$ (adversarial perturbation bound) and the entire spatial robustness profile over varying $\theta$ (range of training augmentation). In this setting, a theoretical proof of even the adversarial robustness vs. accuracy trade-off for varying $\epsilon$ is not known and would be an independent question for future work.

---

### Official Review · Reviewer_h24y · 2021-07-16

**Rating:** 7
**Confidence:** 3

**Summary:**

The work presents a comprehensive discussion on the trade-off between spatial robustness (robust against spatial transformations like rotation, translation, etc) and adversarial robustness (robust against adversarial attacks) in neural network models, by proving the phenomenon theoretically and justifying it empirically. The experiment results show that the model trained with better spatial robustness via equivariant models and larger training augmentation has worse adversarial robustness, and vice versa. To improve both robustness, the authors designed a curriculum learning-based approach where the difficulty of spatial augmentation and adversarial strength is gradually increased during training.

**Limitations And Societal Impact:**

Yes, the authors addressed the limitations and societal impact.

**Main Review:**

- Originality: The work discussed a very interesting phenomenon, which looks counterintuitive at first sight but is actually justifiable. The related works are cited in the introduction section. There is no related work section. I am not familiar with this field, so I cannot comment on if related work is adequately cited.

- Quality: The work is technically sound. The claims are well supported by theoretical analysis and empirical results.

- Clarity: The work is clearly written, and the theoretical analysis and empirical results are well organized. The table and figures are easy to follow. The theoretical results are a little difficult to understand, especially for people like me who are not familiar with the cyclic codewords setting. It would be better to add some figure to show the setup of the theoretical analysis.

- Significance: The results for justifying the trade-off between two types of robustness are clear and significant. The results for the performance of the proposed curriculum learning algorithm (Table 1) are not that clear to me. It seems the proposed method achieves better spatial accuracy than PGD, but worse adversarial accuracy than Aug 180 -> PGD.  So I am not sure if you can claim that the proposed method improves both types of robustness (Abstract).  Moreover, as justified in the first part of the paper, there is always a trade off between the two types, so even with the proposed curriculum learning method, it should not achieve the improvement for both.


Update: Thank the authors for answering to my questions. I keep my rating. I would suggest the authors incorporate the new results into the final version.

**Time Spent Reviewing:**

3

---

> ### Author Response · Authors · 2021-08-10
> **Response to Reviewer h24y**
>
> >Clarity: The work is clearly written, and the theoretical analysis and empirical results are well organized. The table and figures are easy to follow. The theoretical results are a little difficult to understand, especially for people like me who are not familiar with the cyclic codewords setting. It would be better to add some figure to show the setup of the theoretical analysis.
>
> Thank you, we will take up your suggestion and think over potential visual representations of our theoretical results in the final version.
>
> >Significance: The results for justifying the trade-off between two types of robustness are clear and significant. The results for the performance of the proposed curriculum learning algorithm (Table 1) are not that clear to me. It seems the proposed method achieves better spatial accuracy than PGD, but worse adversarial accuracy than Aug 180 -> PGD. So I am not sure if you can claim that the proposed method improves both types of robustness (Abstract). Moreover, as justified in the first part of the paper, there is always a trade off between the two types, so even with the proposed curriculum learning method, it should not achieve the improvement for both.
>
> Fig. 6 and Table 1 show the Pareto-efficiency (the best across two metrics when there is a trade-off between them) of our curriculum learning method CuSP when compared against various natural combinations of training augmentation and adversarial training. We **do not** claim that our method Pareto-dominates the others or improves over them in **both** adversarial and spatial accuracy. Instead, our method is Pareto-optimal when compared against many natural baselines (i.e., none of these natural baselines Pareto-dominate our method), and it gives a recipe to improve adversarial and spatial robustness simultaneously (as opposed to improving mostly one of them, as done by many natural baselines).

---

### Official Review · Reviewer_wu4B · 2021-07-20

**Rating:** 7
**Confidence:** 4

**Summary:**

This paper studies the properties of spatial and adversarial $\epsilon$-bounded robustness of convolutional neural networks. Throughout the paper, the authors focus on __random__ spatial transformations, as opposed to __targeted__ (adversarial) rotations/translations. Their key hypothesis is that spatial robustness and adversarial $\epsilon$-bounded robustness are inimical to each other, and increasing one must be at the cost of decreasing the other.

They derive theoretical results for a specific distribution, and derive upper and lower bounds for the spatial robustness of an adversarially trained network on this distribution. They also show that high accuracy on the original data does not imply high spatial robustness.

Through a set of elaborately designed experiments, they provide empirical evidence for their main hypothesis. Finally, they propose a curriculum-learning based approach to find a pareto-optimal solution that achieves the optimal trade-off between the two types of robustness.

**Limitations And Societal Impact:**

I do not see any possible negative societal impact arising from this paper.

**Main Review:**

**High-level comments :**

The paper is nicely structured, and the experiments are thorough. Overall, it provides compelling evidence for the hypothesis that there is a trade-off between the two notions of robustness. The theoretical results provide support for the main hypothesis, although I have not checked the proof of Theorem 2 in detail.

**Questions :**

1. Do you think a similar trade-off holds for the worst-case spatial robustness? That is, choosing a custom rotation/translation for each image and neural network, using an approach like in [1].

2. It would be nice if you could do one experiment with more than 10 iterations of PGD on CIFAR-10. Unless PGD converges, it is sometimes difficult to draw definitive conclusions from the results.

3. I am slightly confused about the role of $m$ in the theoretical analysis. If all the coordinates (other than $x_0$ are identical), then how does shifting by different amounts correspond to a greater or lesser degree of transformation? For example, couldn’t one just re-number the coordinates without changing anything (modulo the choice of code word)?

**Minor remarks :**

1. Line 63 - “First, we show that spatial robustness on the above distribution is non-trivial...”. This is confusing because at first glance it seems like “the model has spatial robustness to some non-trivial extent”, which is actually the opposite. I would re-word it as “First, we show that achieving a high degree of spatial robustness on the above distribution is non-trivial…”.

2. In Proposition 1, could you qualify the meaning of $j$ in both mentions of $r_j$? For instance, “...$(r_j(X), Y)$ for any $j$ also has accuracy at least 97%.”

3. Could you add a row in Table 1 for a standard model trained without PGD or augmentation?

**References :**

1. Xiao, Chaowei, et al. "Spatially transformed adversarial examples." 6th International Conference on Learning Representations, ICLR 2018. 2018.

**Time Spent Reviewing:**

6

---

> ### Author Response · Authors · 2021-08-10
> **Response to Reviewer wu4B**
>
> >Do you think a similar trade-off holds for the worst-case spatial robustness? That is, choosing a custom rotation/translation for each image and neural network, using an approach like in [1].
> >References : [1] Xiao, Chaowei, et al. "Spatially transformed adversarial examples." 6th International Conference on Learning Representations, ICLR 2018. 2018.
>
> The worst-case spatial robustness is always upper bounded by the average spatial robustness (e.g., Engstrom et al. [12]). Hence, the trade-off we show for the average spatial robustness is stronger and implies a similar trade-off for the worst-case spatial robustness. If the accuracy against the stAdv attack of Xiao et al. [1] is a good proxy for the worst-case spatial robustness against large rotations/translations, then a similar trade-off could also hold in experiments.
>
> >It would be nice if you could do one experiment with more than 10 iterations of PGD on CIFAR-10. Unless PGD converges, it is sometimes difficult to draw definitive conclusions from the results.
>
> We ran the following experiments for Table 1 (row 1 and 2) with PGD with different setting and still obtain similar trends. We checked adversarial accuracy for the following 4 variants of PGD attack with $\epsilon=8/255$ but different no. of steps $k$ and step sizes.
>
> a) $k$ = 10, step size = $2/255$,
> b) $k$ = 20, step size = $2/255$ or $0.0078$,
> c) $k$ = 100, step size = $2/255$ or $0.0078$,
> d) $k$ = 100, step size = $1/255$ or $0.0039$
>
> Our conclusions continue to hold for (a-b-c-d) above (comparing with our results in Table 1).
>
> **[Table 1 row 1] PGD(Aug 0)**
>
> |Std Accuracy(%) | Spatial Accuracy(%) | Adv (PGD) Accuracy(%) (a-b-c-d)|
> |--|--|--|
> |79.60 | 32.51 | 45.05 - 44.75 - 43.26 - 43.27|
>
> **[Table 1 row 2] PGD(Aug 180)**
>
> |Std Accuracy(%) | Spatial Accuracy(%) | Adv (PGD) Accuracy(%) (a-b-c-d)|
> |--|--|--|
> |54.29 | 53.92 | 33.11 - 32.99 - 32.79 - 32.71 |
>
> >I am slightly confused about the role of in the theoretical analysis. If all the coordinates (other than x_0 are identical), then how does shifting by different amounts correspond to a greater or lesser degree of transformation? For example, couldn’t one just re-number the coordinates without changing anything (modulo the choice of code word)?
>
> The other coordinates are not identical. $X_t|Y=y$ is normally distributed as $N(2c_t y/\sqrt{d}, 1)$ and $c_t$ is different for different values of t. See lines 156-157.
>
> >Minor remarks :
> Line 63 - “First, we show that spatial robustness on the above distribution is non-trivial...”. This is confusing because at first glance it seems like “the model has spatial robustness to some non-trivial extent”, which is actually the opposite. I would re-word it as “First, we show that achieving a high degree of spatial robustness on the above distribution is non-trivial…”.
>
> Thank you for the careful reading. We will reword it as per your suggestion.
>
> >In Proposition 1, could you qualify the meaning of j in both mentions of r_j? For instance, “... (r_j(X), Y) for any j also has accuracy at least 97%.”
>
> We meant that the distribution of the rotated instance $(r_j(X), Y)$ also allows high accuracy classifiers of accuracy more than 97%.
>
> >Could you add a row in Table 1 for a standard model trained without PGD or augmentation?
>
> The numbers for StdCNN/VGG16 on CIFAR10 without PGD or augmentation are below. We will include these in the final version.
>
> |Training Method | Adv (PGD) Accuracy(%) | Std Accuracy(%) | Spatial Accuracy(%) |
> |---------------|-------|-------|-------|
> |Natural (Aug 0)| 00.00 | 93.19 | 34.27 |

---

### Official Review · Reviewer_Uu2b · 2021-07-21

**Rating:** 6
**Confidence:** 4

**Summary:**

This paper explores the trade-off between (1) average-case robustness to translation and rotations and (2) adversarial robustness. The paper shows through extensive experiments and a theoretical example that the trade-off occurs, and they show that curriculum learning can lead to a reasonable trade-off that achieves some degree of both types of robustness.

**Ethical Concerns:**

None.

**Limitations And Societal Impact:**

The authors have addressed the societal impact of their work.


**Main Review:**

Overall, the paper is clear and well-written, and has mostly solid supporting experimental evidence. Thus, the quality and clarity of the work is high. On the other hand, the originality and significance is more limited, but I feel that it is still interesting. Thus, I would recommend a weak accept for this paper.

This paper explores the question of whether or not robustness to random translations and rotations (“spatial robustness”) can be achieved while also achieving adversarial robustness, and the authors show in a specific theoretical construction that the answer is “no” -- that is, there exists a distribution on which a classifier of high adversarial robustness necessarily has relatively low spatial robustness. They then show experimental evidence that when optimizing purely for adversarial robustness (via standard PGD-adversarial training), spatial robustness decreases, and vice versa. In particular, they measure metrics that are reasonable, such that all models start at the same value for that metric and performance degradation occurs only after the adversarial epsilon (or the amount of rotation/translation) is increased. The results are presented in a fairly clear and logical manner, and the experiments are relatively comprehensive, covering multiple datasets and a few different models and architectures.

The authors also show that a simple curriculum-learning approach can lead to relatively high adversarial robustness and spatial robustness; I am not too sure of the significance of this, as I do not know if optimizing for this combination of robustness (as opposed to, say, adversarial robustness and common corruption robustness) is particularly useful for any specific downstream task. However, understanding how different types of robustness in general can be mixed with each other is valuable, so I do think it is still interesting.

I do have some more minor questions/clarifications/suggestions for the authors to improve their work.

1) Can you also include evaluation results for adversarial spatial robustness? For example, it might be interesting to know how well the curriculum-learned models perform when faced with adversarial rotations/translations -- does it still work well?
2) Around L105-113: What is an example of a truly invariant model, vs. a model that is invariant for most inputs but not all x?
3) L236 - when you measure adversarial robustness using PGD, are you still using the same number of PGD-attack steps as during train-time? Usually, it is better to use a greater number of attack steps (e.g. 100 steps instead of 10 steps for CIFAR) when evaluating test-time adversarial robustness, as sometimes PGD attacks are still relatively weak without more attack-steps.
4) L244 - in general, are there significant differences between StdCNNs/GCNN’s, or different architectures? From my inspection of Figure 1, the overall trends seem similar.
5) Table 1 - Have you also tried PGD (Aug 0) -> PGD (Aug 180), or Aug 180 -> PGD (Aug 180)? Those both seem like more natural ways to try to achieve both types of robustness, as opposed to completely ignoring one type of robustness during the second phase of training.
6) Figure 6 - it seems that the curriculum learning helps a little bit, but not too much, relative to something simpler like PGD (Aug 180). Could this relatively minor difference just be a result of the learning rate schedule, for example? That is, is it possible that the curriculum learning step is not actually necessary?



------------
Update: The authors responded to my questions and also performed new experiments, which had results that still support their original conclusions (and also, the new experiments seem to get even better numbers than the original experiments in the paper).

Thus, I still believe that the paper should be accepted, and I would encourage the authors to add some of the updated experimental results into their paper for completeness.



**Time Spent Reviewing:**

3

---

> ### Author Response · Authors · 2021-08-10
> **Response to Reviewer Uu2b**
>
> >Can you also include evaluation results for adversarial spatial robustness? For example, it might be interesting to know how well the curriculum-learned models perform when faced with adversarial rotations/translations -- does it still work well?
>
> The adversarial spatial accuracy against worst-case rotations in the $[-\theta, +\theta]$ range is upper-bounded by the average spatial accuracy against random rotations from $[-\theta, +\theta]$ (e.g., Engstrom et al. [12]). Hence, our trade-off for average spatial robustness is stronger and implies a similar trade-off for adversarial spatial robustness. The trade-off is more significant for larger ranges and the spatial accuracy reported in Table 1 is against random rotations from $[-180^{\circ}, +180^{\circ}]$ (see lines 323-325). Existing spatial adversarial attacks [12, 42] use only small rotations up to $\pm 40^{\circ}$ and small translations up to $\pm 3$ pixels. We are not aware of any spatial adversarial attacks that look at the entire range of rotations/translations. We would be happy to evaluate our curriculum-learned models against them, if that adds value.
>
> >Around L105-113: What is an example of a truly invariant model, vs. a model that is invariant for most inputs but not all x?
>
> Group-equivariant CNNs for the rotation group *p*4 (ref. [8]) are invariant to rotations that are multiples of $90^{\circ}$ by construction but they are not truly invariant to all rotations.
>
> >L236 - when you measure adversarial robustness using PGD, are you still using the same number of PGD-attack steps as during train-time? Usually, it is better to use a greater number of attack steps (e.g. 100 steps instead of 10 steps for CIFAR) when evaluating test-time adversarial robustness, as sometimes PGD attacks are still relatively weak without more attack-steps.
>
> We checked adversarial accuracy for the following 4 variants of PGD attack with $\epsilon=8/255$ but different no. of steps $k$ and step sizes.
>
> a) $k$ = 10, step size = $2/255$,
> b) $k$ = 20, step size = $2/255$ or $0.0078$,
> c) $k$ = 100, step size = $2/255$ or $0.0078$,
> d) $k$ = 100, step size = $1/255$ or $0.0039$
>
> Our conclusions continue to hold for (a-b-c-d) above.
>
> **[Table 1 row 1] PGD(Aug 0)**
>
> |Std Accuracy(%) | Spatial Accuracy(%) | Adv (PGD) Accuracy(%) (a-b-c-d)|
> |--|--|--|
> |79.60 | 32.51 | 45.05 - 44.75 - 43.26 - 43.27|
>
> **[Table 1 row 2] PGD(Aug 180)**
>
> |Std Accuracy(%) | Spatial Accuracy(%) | Adv (PGD) Accuracy(%) (a-b-c-d)|
> |--|--|--|
> |54.29 | 53.92 | 33.11 - 32.99 - 32.79 - 32.71 |
>
>
> >L244 - in general, are there significant differences between StdCNNs/GCNN’s, or different architectures? From my inspection of Figure 1, the overall trends seem similar.
>
> Yes, the overall trends are similar even though StdCNNs and GCNNs are equivariant to different groups of transformations by design.
>
> >Table 1 - Have you also tried PGD (Aug 0) -> PGD (Aug 180), or Aug 180 -> PGD (Aug 180)? Those both seem like more natural ways to try to achieve both types of robustness, as opposed to completely ignoring one type of robustness during the second phase of training.
>
> Those are great suggestions for the completeness of Table 1. The two methods you mentioned give the following results.
> PGD (Aug 0) -> PGD (Aug 180)
>
> | Adv (PGD) Accuracy(%) | Std Accuracy(%) | Spatial Accuracy(%) |
> |-------|-------|-------|
> | 35.50 | 59.15 | 57.99 |
>
> Aug 180 -> PGD (Aug 180)
>
> | Adv (PGD) Accuracy(%) | Std Accuracy(%) | Spatial Accuracy(%) |
> |-------|-------|-------|
> |36.25  | 60.32 | 58.89 |
>
> Following them with CuSP gives even better results, as shown below!
>
> PGD (Aug 0) -> CuSP(PGD,120-150-180,{$\frac{2}{255},\frac{4}{255},\frac{8}{255}$})
>
> | Adv (PGD) Accuracy(%) | Std Accuracy(%) | Spatial Accuracy(%) |
> |-------|-------|-------|
> |37.20  | 66.93 | 62.63 |
>
> Aug 180 -> CuSP(PGD,120-150-180,{$\frac{2}{255},\frac{4}{255},\frac{8}{255}$})
>
> | Adv (PGD) Accuracy(%) | Std Accuracy(%) | Spatial Accuracy(%) |
> |-------|-------|-------|
> |37.11  | 66.68 | 63.15 |
>
> >Figure 6 - it seems that the curriculum learning helps a little bit, but not too much, relative to something simpler like PGD (Aug 180). Could this relatively minor difference just be a result of the learning rate schedule, for example? That is, is it possible that the curriculum learning step is not actually necessary?
>
> Below are results with a few different learning rate schedules, in addition to Table 6 in the paper.
>
> Learning rate schedule 40-80:
>
> | Adv (PGD) Accuracy(%) | Std Accuracy(%) | Spatial Accuracy(%) |
> |-------|-------|-------|
> |34.00  | 57.40 | 56.40 |
>
> Learning rate schedule 50-100:
>
> | Adv (PGD) Accuracy(%) | Std Accuracy(%) | Spatial Accuracy(%) |
> |-------|-------|-------|
> |34.17  | 57.48 | 56.97 |
>
> Even the simple version of CuSP in Table 6 under a similar setting gives a Pareto improvement over PGD (Aug 180) of at least 0.5 - 5.0 - 5.0 in Adv (PGD) Accuracy(%) - Std Accuracy(%) - Spatial Accuracy(%), respectively. This observation and our response to your previous question illustrate that the proposed curriculum learning strategy offers a definite advantage and Pareto-efficiency over other first-cut solutions.

---

> > ### Comment · Reviewer_Uu2b · 2021-08-26
> > **Thank you for your response**
> >
> > The authors responded to my questions and also performed new experiments, which had results that still support their original conclusions (and also, the new experiments seem to get even better numbers than the original experiments in the paper).
> >
> > Thus, I still believe that the paper should be accepted, and I would encourage the authors to add some of the updated experimental results into their paper for completeness.

---

### Author Response · Authors · 2021-08-23
**Request for acknowledgement of the rebuttal**

We thank all the reviewers for the careful reading and thoughtful questions.
We have addressed all the questions raised by the reviewers in the individual responses below. We request the reviewers to go through the responses; if there are any further questions or concerns, we'd be happy to answer them. We hope we have adequately answered your questions, and that the work's impact and results are better highlighted with our responses.

---

### Decision · Program_Chairs · 2021-09-27

**Decision:**

Accept (Poster)

**Comment:**

This work studies tradeoffs between robustness to translations (spatial robustness) and robustness to worst case lp-perturbations (adversarial robustness). First the authors mathematically construct distributions for which tradeoffs between these two forms of robustness provably exist. Then the authors present a series of empirical studies showing that the tradeoff occurs in practice, and methods towards achieving pareto optimality. Reviewers were overall supportive of acceptance, noting that the empirical experiments presented were particularly interesting and well executed. There was a long discussion during the rebuttal period regarding Theorem 2. For the distribution in question, a tradeoff between spatial and adversarial robustness follows as a direct consequence from a tradeoff between adversarial robustness and standard accuracy (which has been shown already theoretically for some distributions in prior work). Reviewers agreed that the original text did not make it clear that this weaker bound followed from the standard accuracy - robustness tradeoff, or how Theorem 2 strengthened upon this weaker bound. However, even with this issue reviewers agreed that the experimental results were arguably interesting enough to warrant publication. After follow up discussions between the AC and the authors, the authors have agreed to rework this section to make the weaker bound explicit (and better discussion on how it follow from  the standard accuracy / adv robustness tradeoff) while highlighting how their construction strengthens this result. With these changes in mind, the paper seems ready for publication.